# Comprehensive cell type decomposition of circulating cell-free DNA with CelFiE

Christa Caggiano[1,2], Barbara Celona [3], Fleur Garton[4], Joel Mefford[2], Brian L. Black [3,5], Robert Henderson[6], Catherine Lomen-Hoerth[7], Andrew Dahl [8] & Noah Zaitlen [2,9✉]

Circulating cell-free DNA (cfDNA) in the bloodstream originates from dying cells and is a promising noninvasive biomarker for cell death. Here, we propose an algorithm, CelFiE, to accurately estimate the relative abundances of cell types and tissues contributing to cfDNA from epigenetic cfDNA sequencing. In contrast to previous work, CelFiE accommodates low coverage data, does not require CpG site curation, and estimates contributions from multiple unknown cell types that are not available in external reference data. In simulations, CelFiE accurately estimates known and unknown cell type proportions from low coverage and noisy cfDNA mixtures, including from cell types composing less than 1% of the total mixture. When used in two clinically-relevant situations, CelFiE correctly estimates a large placenta component in pregnant women, and an elevated skeletal muscle component in amyotrophic lateral sclerosis (ALS) patients, consistent with the occurrence of muscle wasting typical in these patients. Together, these results show how CelFiE could be a useful tool for biomarker discovery and monitoring the progression of degenerative disease.

[1] Bioinformatics Interdepartmental Program, University of California Los Angeles, Los Angeles, CA, USA. [2] Department of Neurology, University of California Los Angeles, Los Angeles, CA, USA. [3] Cardiovascular Research Institute, University of California San Francisco, San Francisco, CA, USA. [4] Institute for Molecular Biosciences, University of Queensland, Brisbane, Australia. [5] Department of Biochemistry and Biophysics, University of California, San Francisco, USA. [6] Department of Neurology, Royal Brisbane and Women's Hospital, Brisbane, Australia. [7] Department of Neurology, University of California San Francisco, San Francisco, CA, USA. [8] Section of Genetic Medicine, Department of Medicine, University of Chicago, Chicago, IL, USA. [9] Department of Computational Medicine, University of California Los Angeles, Los Angeles, CA, USA. ✉email: nzaitlen@g.ucla.edu

Cells die at different rates as a function of disease state, age, environmental exposure, and behavior[1,2]. A quantifiable indication of cell death could facilitate disease diagnosis and prognosis, prioritize patients for admission into clinical trials, and improve evaluation of treatment efficacy and disease progression[3–6]. Circulating cell-free DNA (cfDNA) is a promising candidate biomarker as it is released into the bloodstream after cell death[7–9]. In healthy individuals, cfDNA in the blood arises from normal cell turnover, but in individuals with a disease, cfDNA can come from illness-specific cell death[10]. As a result, cfDNA levels have been shown to be elevated in individuals with cancer, autoimmune diseases, transplantation responses, and trauma[11–14]. CfDNA has also become the clinical standard for noninvasive prenatal testing[15], and many companies and research groups are sequencing cfDNA to identify the presence of somatic mutations related to tumors[16–18].

To understand what drives the increased presence of cfDNA in people with disease, this work focuses on the decomposition of cfDNA in blood into its cell types of origin. While each germline cell has nearly the same DNA sequence, DNA methylation is cell type specific[19], and there is a rich literature of complex tissue decomposition approaches using DNA methylation[20–23]. Recent work has attempted to use cfDNA methylation patterns to decompose tissues of origin for cfDNA[24–27]. These approaches, however, do not address some of the unique challenges of cfDNA. Previous work was designed for reference and input data from methylation chips, which are high coverage and have relatively low noise. Since cfDNA is only present in the blood in small amounts, an onerous amount of blood must be extracted from a patient to get the required amount of input DNA for methylation chips, which may not be practical for clinical use[28]. Other technologies and methods focus on sensitive detection of specific tissues or cancer sites[29–31]. While increasingly powerful, these approaches cannot provide biomarker discovery or comprehensive decomposition of constitutive cell types. In this work, we used whole genome bisulfite sequencing (WGBS) to assess the methylation of cfDNA. Unlike methylation arrays that target specific genomic locations, WGBS covers the entire genome, typically resulting in lower coverage per site, and increased noise relative to array data. Thus, WGBS presents computational challenges for decomposition of methylation data as current computational methods are ill-equipped to handle such noise in either the reference or input. Previous methods are also limited by which DNA methylation sites (CpGs) are chosen. Methylation arrays survey a limited number of CpGs, which may not be maximally informative of cell type. Some approaches also rely on selecting a set of CpGs designed for a particular dataset[24,26,32]. While curated site selection is useful for specific biological queries, it can cause bias when generalized to other settings or diseases. Choosing which sites to include in decomposition can substantially influence which cell types are predicted because different sites are informative for different cell types. Another important limitation of previous cfDNA decomposition methods is that the results are restricted to the cell types included in the reference panel. However, as there are many thousands of cell types throughout the body, it is currently impossible to incorporate them into a reference panel. Thus, the specific choice of reference cell types will lead to biases in the decomposition results.

In this work, we develop an efficient expectation maximization (EM) algorithm, CelFiE (CELl Free DNA Estimation via expectation-maximization) for cfDNA decomposition that allows for low coverage and noisy data and apply it in a range of challenging real world scenarios. CelFiE can estimate unknown cell types not included in a reference panel and is not dependent on curated input methylation sites. We show in realistic simulations that CelFiE can accurately estimate known and unknown cell types, even at low coverage and with relatively few sites, and can detect rare cell types that contribute to only a small fraction of the total cfDNA. Decomposition of real WGBS complex mixtures demonstrates that CelFiE is robust to several violations of the model assumptions. Specifically, the real data contain correlations across regions and between cell types, read counts with heavy-tailed distributions, and reference samples that are heterogeneous mixtures of many cell types. Additionally, we develop an approach for unbiased CpG methylation site selection for use in the decomposition algorithm.

We apply CelFiE to two cfDNA data sets. First, we examined the positive control of cfDNA extracted from pregnant and nonpregnant women. We observe a significant placental component in the decomposition estimates only from pregnant women, providing validation for CelFiE. We then applied CelFiE to cfDNA from amyotrophic lateral sclerosis (ALS) patients and age-matched controls. Currently, there are no established circulating biomarkers for ALS. As a result, it is difficult to monitor disease progression and efficiently evaluate treatment response[33]. cfDNA provides an opportunity to measure cell death in ALS that could fill these gaps. We find a significantly elevated skeletal muscle component in ALS patients. This observation, along with the successful decomposition of cfDNA from pregnant women, demonstrates that CelFiE has the potential for broad translational utility in understanding the biology of cell death, and in applications such as quantitative biomarker discovery, or in the noninvasive monitoring and diagnosis of disease.

## Results

**CelFiE overview**. CelFiE estimates the contribution of various cell types to the cfDNA of an individual via an EM optimization algorithm. The input to CelFiE is WGBS reference data consisting of $T$ total cell types and WGBS cfDNA samples for $N$ total individuals. Its output is the proportion of the reference cell types that make up each individual's cfDNA, such that the proportion of all $T$ cell types sums to one for each individual. Notably, an arbitrary number of cell types can be missing, which addresses potential biases arising from estimating the proportions of cell types from a restricted reference panel. CelFiE also estimates the methylation values for each of the cell types included in the reference, which accommodates the currently noisy and low-coverage reference data sets. These developments are facilitated by CelFiE's EM algorithm, which is a flexible framework for parameter estimation, even when there is missing data. Complete details on CelFiE can be found in the "Methods" section and Supplementary Note.

**Evaluation using simulated cfDNA mixtures**. We began by simulating cfDNA mixtures informed by realistic sequencing conditions and comparing the results of CelFiE and other decomposition tools. First, we compared CelFiE to a least-squares regression optimization method. Least-squares regression is a popular choice for decomposition problems, but is not guaranteed to produce an estimate of cell type proportions that sums to one. To compare CelFiE to a constrained optimization method, we implemented a second optimization method referred to here as the "projection method". In this approach, we computed the projection of the cell-type proportion estimates onto the L1-ball[34], which constrained the estimates of cell-type proportions to lie on the probability simplex and thus, sum to one. Furthermore, in our projection approach, we optimize a binomial log-likelihood that is parameterized by the number of methylated and unmethylated reads. By accounting for read data, this method is a more direct comparison to CelFiE (see the "Methods" section for implementation details).

We also compared CelFiE to a previously published cfDNA decomposition tool, MethAtlas[25]. Unlike CelFiE, which explicitly

models WGBS reads, MethAtlas is designed to decompose methylation array data. MethAtlas also does not model missing data or estimate the methylation values for the reference cell types. Briefly, it optimizes $\|Y\alpha - \beta\|$ using nonnegative least squares constrained by $\alpha \geq 0$, where $Y$ is a reference matrix of array data, $\beta$ is the observed cfDNA methylation measured on an array, and $\alpha$ is the cell type proportions vector that is being solved for. While MethAtlas is not designed for low read count data and thus not directly analogous to CelFiE, it is, to the best of our knowledge, the only other cfDNA decomposition algorithm that allows the inclusion of arbitrary input sites and does not restrict to specific cell types in the reference data.

MethAtlas provides a comprehensive reference matrix, composed of 25 tissues and cell types, over ~6000 CpG loci[35]. To ensure a fair comparison, we simulated data that matched the size of this reference data with 25 cell types and 6000 CpGs. The true methylation proportion of each CpG was drawn independently from a uniform distribution, so that the methylation of each CpG was between 0% and 100%. The choice of a uniform distribution allowed for variability across cell types for a given CpG. To characterize the decomposition performance of CelFiE across both rare and abundant cell types, we defined the true cell type proportion vector as $(1, ..., T)/(0.0ptT + 12)$, where $T = 25$ is the number of cell types truly in the mixture.

For CelFiE and projection method, the input data were the number of methylated reads and read depth at each site. The reference read depths were drawn independently from a Poisson distribution centered at 10, a relatively low sequencing depth for a WGBS experiment[36]. The number of methylated reads for a given CpG in each of the 25 cell types was drawn from a binomial distribution, where the probability of success was the true methylation value in that cell type, and the number of trials was the read depth at that locus. cfDNA read depths for each CpG were simulated from a Poisson distribution centered at 10, and then the reads for each CpG were assigned to originate from a cell type based on the cell type proportion vector for the cfDNA mixture. A read was determined to be either methylated or unmethylated given the true methylation proportion in that read's cell type of origin at that CpG. Since MethAtlas and least-squares regression do not take read counts as input, we calculated the methylation proportion for a CpG by dividing the methylated reads by the depth at that locus. While these methods were not designed for read count data, by doing this we were able to compare MethAtlas, least-squares regression, and CelFiE on the same data. Additionally, to compare the least-squares regression estimates to the proportions produced by the other methods, we divided the vector of estimates produced by least-squares regression by its sum. In total, we performed 50 independent simulations for CelFiE and all comparison methods. Below, we consider additional simulations from real data, which are free from the distributional assumptions above.

CelFiE performed better than MethAtlas at these low-read depths (Fig. 1). Per each simulation, we calculated Pearson's correlation between the true cell type proportion vector and the estimated proportions vector. For MethAtlas, the mean $r^2$ across replicates was $0.59 \pm 0.17$, while CelFiE's mean $r^2$ was $0.96 \pm 0.01$. As expected, CelFiE also performed better than linear least-squares regression, which had a mean $r^2$ of $0.73 \pm 0.11$ (Fig. S1A). CelFiE and the projection optimization method (mean $r^2 = 0.95 \pm 0.02$) performed similarly under these conditions (Fig. S1B). However, a major limitation of our projection optimization method is that, unlike CelFiE, it is unable to estimate missing cell types, which we discuss further below.

To further characterize the properties of CelFiE, we varied the number of CpGs (100, 1000, and 10,000), which represented conditions with varying amounts of information about cell type.

We then focused on a single cell type and varied its proportion between 0% and 100%. In total, we simulated 10 cell types, where one cell type was fixed. The remaining 9 additional cell type proportions were drawn from an independent uniform distribution and then normalized so that all proportions sum to one. Data were simulated for 1 individual with 50 independent simulations.

Performance was assessed by calculating the Pearson's correlation between the estimated cell-type proportions and the true proportions for 50 replicates. We found that as the number of sites increased, the ability of CelFiE to accurately decompose the cfDNA mixtures improved (Fig. 2a), especially for less abundant cell types. We further characterized the performance of CelFiE by calculating the correlation between the estimated methylation proportions of the fixed cell type with the true methylation proportions when the reference and input data were at 1×, 5×, 10×, or 100× coverage (Fig. 2b). At the very low depth of 1×, the mean Pearson's correlation was $r^2 = 0.45 \pm 0.09$, which increased substantially at 5× coverage to $r^2 = 0.83 \pm 0.03$. As the sequencing depth increased, the correlation continued to increase.

Next, we examined the performance of CelFiE when two cell types with highly correlated methylation values were included in the reference panel, since many real cell types share substantial architecture with each other. We generated simulated methylation proportions for the two cell types with a Pearson's correlation between 0 and 1 at 100× depth and ran CelFiE for mixtures of 1000 CpG sites. When the cell types are very correlated, we found that CelFiE is unable to distinguish between the two cell types. As the cell types become less related, CelFiE improved in its ability to disambiguate the two cell types (Fig. S2). We note, however, that CelFiE accurately estimates the sum of the two cell types, even when they are perfectly correlated.

**Detection of differences between groups.** Previous work suggests that a large portion of cfDNA originates from white blood cells[24]. This implies that a non-hematopoietic cell type of clinical significance may only be present in a population of interest at a low proportion in the mixture. To assess the ability of CelFiE to estimate rare cell types, we simulated data to resemble a small case-control study of 10 total individuals. Five individuals with a low proportion of a single cell type (0.1%, 0.5%, 1%, or 5%) were simulated to represent the cases. The remaining 5 individuals were simulated to have 0% of that cell type, representing the controls. To understand how CelFiE's ability to estimate rare cell types changes as a function of sequencing depth, we simulated input and reference reads at 5×, 10×, 100×, and 1000× coverage for 1000 CpGs. We ran CelFiE jointly on all 10 individuals to prevent bias and assessed whether CelFiE can meaningfully discriminate between the two groups.

We plotted the CelFiE estimates for individuals whose cfDNA mixtures do and do not have that rare cell type (Fig. 3A–D). We found that as both the depth and the cell-type proportion increased, CelFiE's ability to distinguish between the two groups improved. A grouped $t$-test was used to assess whether CelFiE is estimating a significant difference between the groups. At a depth of 5×, CelFiE was only able to distinguish between the groups of the most abundant fixed cell type, 5%, with an average estimate of $0.041 \pm 0.018$ in the group with the cell type and $2.51 \times 10^{-3} \pm 4.71 \times 10^{-3}$ in the group without. Despite the estimates being slightly underestimated, this difference was significantly different between the groups ($p = 4.8 \times 10^{-8}$), suggesting that CelFiE may have utility in detecting differences between groups even at extremely low depths. As the depth increased to 1000×, CelFiE significantly differentiated between all four fixed percentages ($p < 0.001$) and the estimates became more confident. We found that

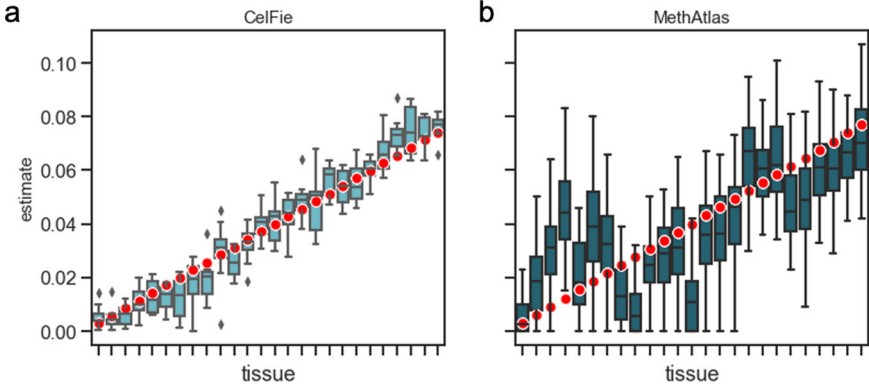

**Fig. 1 Decomposition of simulated cfDNA mixtures by CelFiE (A) and MethAtlas (B).** 50 replications for a single simulated individual were performed, and the estimated mixing proportions were plotted (light blue and dark blue boxes, respectively). The red dots indicate the true cell type proportion for each simulated tissue. The center line of the box indicates the mean, the outer edges of the box indicate the upper and lower quartiles, and the whiskers indicate the maxima and minima of the distribution.

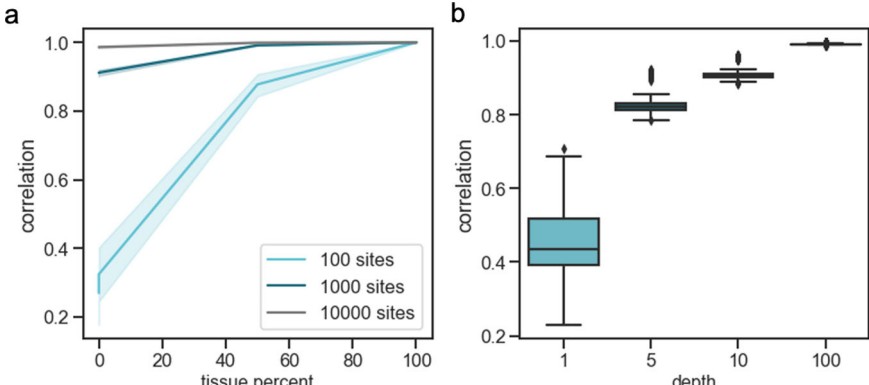

**Fig. 2 The performance of CelFiE on simulated mixtures.** First, a cell type is fixed at a proportion between 0% and 100%, and reads are simulated for 100 (light blue line), 1000 (dark blue line), and 10,000 (black line) CpG sites at 10× depth (**a**). The Pearson's correlation between the true and estimated cell-type proportion is plotted. Solid lines indicate the mean and the shading around the line indicates a 95% confidence interval. On (**b**) the average Pearson's correlation between the true methylation values for the fixed tissues and the CelFiE estimated methylation values for 1000 sites simulated with 1×, 5×, 10×, and 100× depths (light blue boxes). The center of the boxplot indicates the mean of the distribution, the edges of the box indicate the upper and lower quartiles, and the edge of the whiskers indicate the maxima and minima of the distribution. Data is shown for 50 independent simulations of one individual.

as we continued to increase the depth, CelFiE was able to detect arbitrarily small differences between the groups (at 10,000× and 0.01%, $p = 8.32 \times 10^{-9}$). In practice, however, the ability of CelFiE to detect these minute differences is limited by biological and technical constraints, such as the amount of cfDNA in blood or DNA degraded by bisulfite conversion. Nonetheless, these results demonstrate that CelFiE can accurately estimate cell types of relatively rare abundance when the read depth is high.

**Unknown cell types**. We then turned to understand the behavior of CelFiE when estimating unknown cell types. To accomplish this, we simulated data with low read counts, creating reference and cfDNA reads for 1000 CpGs at 10× depth, as in previous simulations. We simulated $t = 10$ cell types with one unknown cell type excluded from the reference data. We began by simulating a missing component that was relatively large. Its proportion, $\alpha_{unknown}$, was drawn as $\alpha_{unknown} \sim \mathcal{N}(0.2, 0.1)$ and truncated to be between 0 and 1. The remaining cell type proportions of the known cell types were drawn from a uniform distribution and all proportions were normalized to sum to 1. We then simulated cfDNA reads for 10, 50, 100, 500, and 1000 individuals. Note that the problem is not identifiable when the number of individuals is smaller than the number of unknowns.

The mean squared error (MSE) was calculated between the estimated unknown proportion and the true simulated proportion. As the number of people included in the decomposition was increased, the performance of CelFiE improved (Fig. 4a).

We next considered mixtures with two unknown cell types, one that was relatively large and one that was relatively small. For each person, the first unknown proportion, $\alpha_{unknown1}$, was drawn from $\alpha_{unknown1} \sim \mathcal{N}(0.2, 0.1)$, and the second unknown was drawn from $\alpha_{unknown2} \sim \mathcal{N}(0.1, 0.1)$. The proportions of the remaining cell types were simulated as above. Since the inferred CelFiE labels are not identified (i.e., CelFiE's estimated $\alpha_{unknown1}$ can correspond to either missing reference cell type 1 or 2), we assigned the unknowns by examining the estimated methylation fractions of each CpG. We estimated the correlation between the true and unknown methylation fractions and assigned the unknown to the true cell type with the highest correlation. After assigning the unknowns, we calculated the MSE between the true proportion and the estimated proportion. Furthermore, we calculated the Pearson's correlation between the true and estimated methylation fractions for each unknown (Fig. S3). We observed that more individuals are needed to accurately estimate the unknown components when an additional unknown was added (Fig. 4b). We also noted the presence of outliers in the

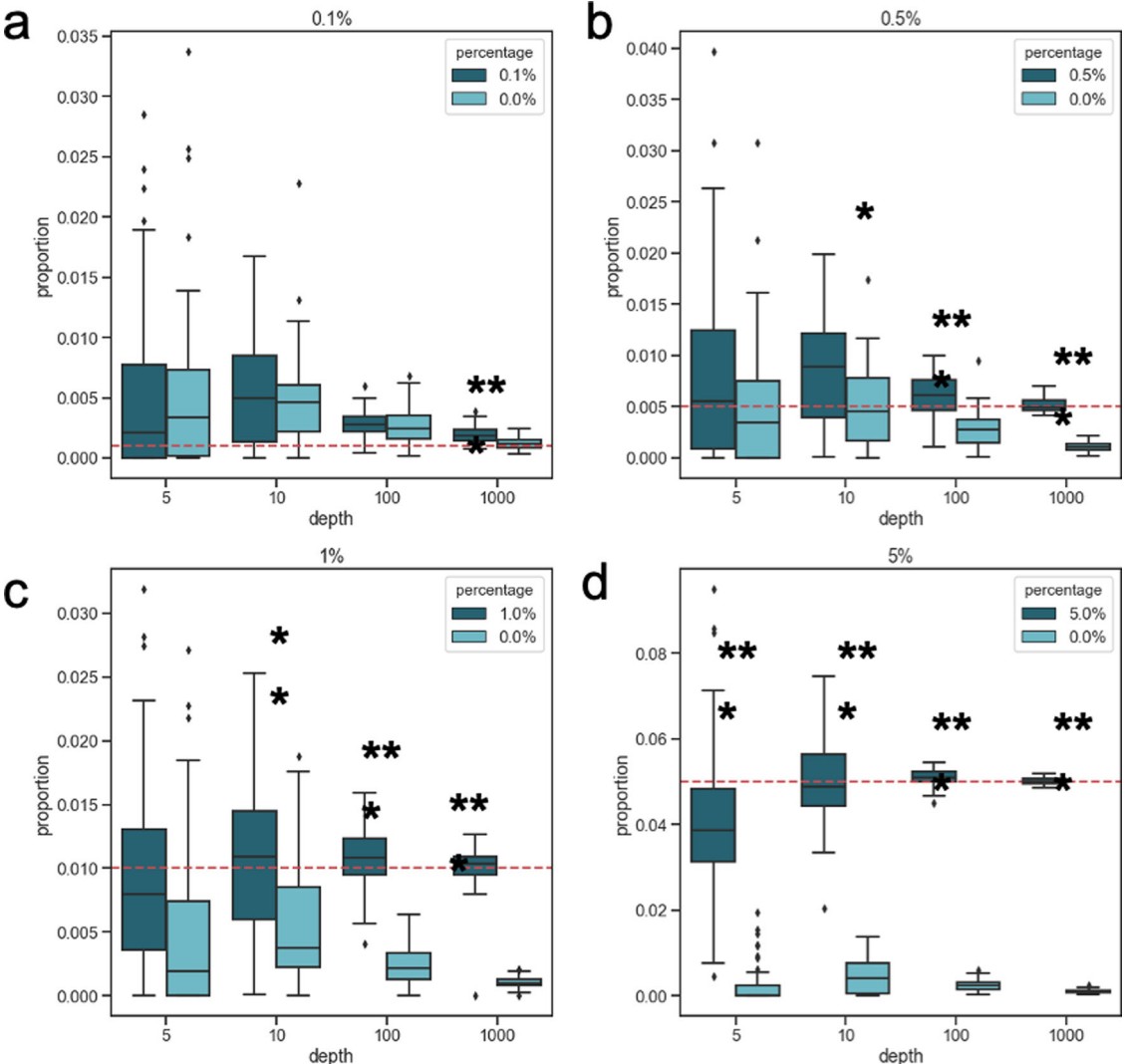

**Fig. 3 Cell type proportion estimates for $n = 5$ simulated individuals (dark blue boxes) with a cell type of interest and $n = 5$ individuals without that cell type (light blue boxes).** Cell type proportions are simulated at (**a**) 0.1% (two-sided grouped $t$-test; 5×: n.s., 10×: n.s, 100×: n.s., 1000×: $p = 3.5 \times 10^{-5}$), (**b**) 0.5% (two-sided grouped $t$-test; 5×: n.s., 10×: $p = 0.013$, 100×: $2.1 \times 10^{-6}$, 1000×: $p = 5.7 \times 10^{-11}$), (**c**) 1% (two-sided grouped $t$-test; 5×: n.s., 10×: $p = 1.5 \times 10^{-3}$, 100×: $2.8 \times 10^{-9}$, 1000×: $p = 4.3 \times 10^{-12}$), or (**d**) 5% (two-sided grouped $t$-test; 5×: $4.8 \times 10^{-8}$, 10×: $p = 5.4 \times 10^{-9}$, 100×: $1.8 \times 10^{-14}$, 1000×: $p < 2.0 \times 10^{-16}$). The true fixed percentage of the cases is indicated by a red dotted line. Significant differences between the groups are indicated by *($p < 0.05$), **($p < 0.01$), and ***($p < 0.001$). The centerline of the box indicates the mean, the outer edges of the box indicate the upper and lower quartiles, and the whiskers indicate the maxima and minima of the distribution. Data is shown for 50 independent simulations.

estimates, which was likely due to differences in the simulated data that were randomly drawn in each replicate of our experiment.

We next examined how decomposition estimates are biased when there is a missing cell type, but no unknown is estimated. We generated simulated mixtures as above, for 1000 CpGs and 10 cell types truly in the reference, and for 100 people at 100× depth. CelFiE was ran twice: once when the missing cell type was the highest tissue in the mixture (~20%) and secondly, when the missing cell type was approximately the average of all cell types contained in the mixture (~10%). (Fig. S5). To measure the bias of the estimates, we calculated the percent difference, defined as the true cell type proportion minus the estimate, divided by the true proportion. When the missing cell type was high, the average percent difference across all tissues was $0.32 \pm 0.86$. This meant that on average, without estimating the unknown, CelFiE produced cell type proportion estimates that were 32% higher than the truth. Likewise, when the missing cell type was lower, the

average percent difference decreased to $0.21 \pm 0.69$, likely because there was less missing signal to be distributed across the cell types actually estimated. When there was an unknown included in CelFiE, the overestimate on average, decreased to $-0.02 \pm 0.62$ and $-0.11 \pm 0.40$, respectively. This result indicated that the larger the missing cell type, the more biased the cfDNA decomposition estimates will be without an unknown component, which may demonstrate the utility of CelFiE.

CelFiE's ability to accurately estimate unknowns contrasts with previous cfDNA decomposition methods, which can only estimate proportions of cell types in the reference. This creates a bias in the decomposition that can be addressed with CelFiE. Specifically, if we simulate cfDNA mixtures with a cell type excluded from the reference as above and run MethAtlas, it will produce biased estimates. On average, these estimates had an average percent difference that was $29 \pm 68\%$ larger than the true proportions (excluding the missing cell type, which was not estimated). We found similar biases in our least-squares

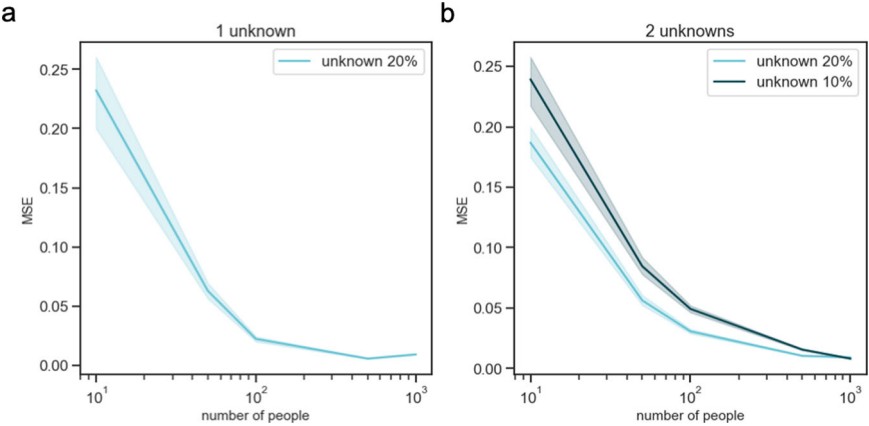

**Fig. 4 Decomposition results for 50 independent simulations of cfDNA mixtures with missing cell types in the reference.** We simulate cfDNA for 10, 50, 100, 500, and 1000 people, and exclude one cell type truly in the mixture at 20% (light blue) (**a**) or two cell types (**b**), one in the mixture at a mean proportion of 20% (light blue), and the other at 10% (dark blue). We calculate the MSE between the true unknown proportion and the CelFiE estimate for 50 simulation experiments. The 95% confidence interval is indicated by the light and dark blue shading.

regression method, which on average, overestimated by $29 \pm 20\%$, and in our projection method, which on average, overestimated by $17 \pm 57\%$ (Fig. S4). The difference in performance between CelFiE and comparison methods is more similar at high read depths and when all cell types are known (Fig. S6).

**Performance on WGBS cfDNA mixtures.** We next considered simulated mixtures made from real WGBS data, which are substantially more complex and violate several assumptions of the CelFiE algorithm. In particular, the reference data contain tissues composed of multiple cell types, CpGs are correlated locally across genomic regions and between cell types, and read counts have heavy-tailed distributions reflecting true biological and technical heterogeneity across sites. Therefore, to examine how robust CelFiE is to these complications, we used biological replicates for 10 WGBS data sets (small intestine, pancreas, monocytes, stomach, tibial nerve, macrophages, memory B cells, adipose, neutrophils, and CD4+ T cells), downloaded from the ENCODE and BLUEPRINT projects[37–40]. In all experiments, we chose to include tissues to see if their complex cell type mixtures might contribute to decomposition errors. One set of WGBS biological replicates was assigned to make up the cfDNA mixtures; the other was assigned to the reference matrix.

Since roughly 80% of CpG sites in the human genome do not vary between cell types[41], randomly selected CpGs will contain mostly uninformative loci for cell-type decomposition. A reference panel that contains too many uninformative CpGs will reduce the performance of a decomposition algorithm. To demonstrate this, we simulated data for 100, 1000, and 10,000 CpGs, where the true methylation values for 10 cell types were drawn from a normal distribution centered on 0.5. The variance across cell types was chosen to be between 0.01 and 1. The lower the variance, the less informative a CpG would be for cell type status. A cfDNA mixture for one individual and no missing cell types was simulated. The results of this experiment indicated that as the variance increased, CelFiE's ability to decompose the mixtures also increased (Fig. S7). Therefore, to limit uninformative CpGs included in our analysis, we developed a method for choosing a set of unbiased informative CpGs in real data, which we called tissue informative markers (TIMs) (see the "Methods" section). We selected 100 TIMs per WGBS sample for use in these simulations, excluding common variants with a minor allele frequency >1%[42]. Selecting TIMs improved performance in CelFiE decomposition (Fig. S8). Furthermore, because DNA methylation of nearby CpGs are correlated[43], we combined information from proximal CpG sites 250bp upstream and downstream of each TIM (see the "Methods" section). These combined TIM regions improved the decomposition over single CpGs (Fig. S9). We simulated cell type proportions 50 times for 100 people. The proportion of CD4+ T cells was drawn from a normal distribution centered around 20% and the proportion of small intestine was centered around 10%. The remaining cell types proportions were drawn per person from a random uniform distribution.

We first assessed CelFiE's performance on WGBS samples without any cell type missing from the reference panel (Fig. 5a). Despite the complexity of the data, we found that CelFiE still performed well. The average Pearson's correlation between the estimated cell type proportions and true cell type proportions was $r^2 = 0.83 \pm 0.16$. The average Pearson's correlation of the estimated methylation values and the true methylation values was similarly high, with an average $r^2$ of $0.96 \pm 0.01$ (Fig. S10A). For comparison, we adapted MethAtlas for whole-genome data. We used our selected TIMs and converted the read counts to proportions. Pearson's correlation between the estimated methylation proportions and true proportions for MethAtlas was lower than that of CelFiE, $0.43 \pm 0.24$, which further illustrated that MethAtlas is not suitable for noisy read count data.

Next, we investigated CelFiE's ability to estimate mixtures with a substantial unknown component. We first masked only the most abundant cell type from the reference, the CD4+ T cell sample. Using the same true cell type proportions as in the simulations with no missing samples, we performed 50 simulations with 100 people (Fig. 5b). The correlation between the estimated and true cell type proportions decreased only slightly in the case of no missing data, $r^2 = 0.8 \pm 0.16$, and we found that the correlation to the true methylation values was still high, with an average Pearson's $r^2 = 0.96 \pm 0.01$ across all cell types (Fig. S10B). Subsequently, we masked two reference samples, CD4+ T cell and small intestine, from the reference panel. The true CD4+ T cell proportion was still centered around 20%, while the small intestine was centered around 10%. We found that CelFiE's ability to successfully decompose a complex mixture decreased when there are two missing cell types (Fig. 5c). However, the estimated correlation to the true WGBS methylation values remained high, with an average Pearson's $r^2 = 0.95 \pm 0.04$ (Figs. 5c and S10C).

To further validate CelFiE's ability to estimate missing cell types, we assessed how similar the learned methylation proportions for the missing cell types are to the true methylation

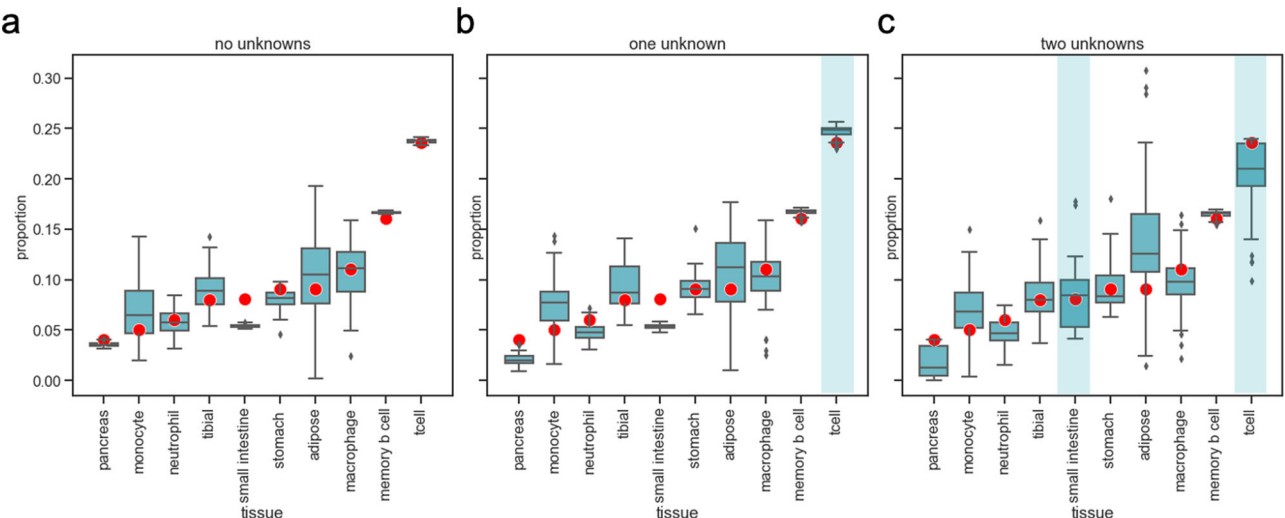

**Fig. 5 CelFiE cell type proportion estimates for a randomly selected individual's real WGBS cfDNA over 50 simulation experiments.** The blue boxes represent estimates of the true cell type composition (red dots) for 100 individuals in 50 simulation experiments in the scenario where there are no missing cell types (**a**), when CD4+ T cells are a missing cell type (indicated by blue shading) (**b**) and when CD4+ T cell and small intestine are both missing (**c**). The center line of the boxplot indicates the mean, the outer edges of the box indicate the upper and lower quartiles, and the whiskers indicate the maxima and minima of the distribution.

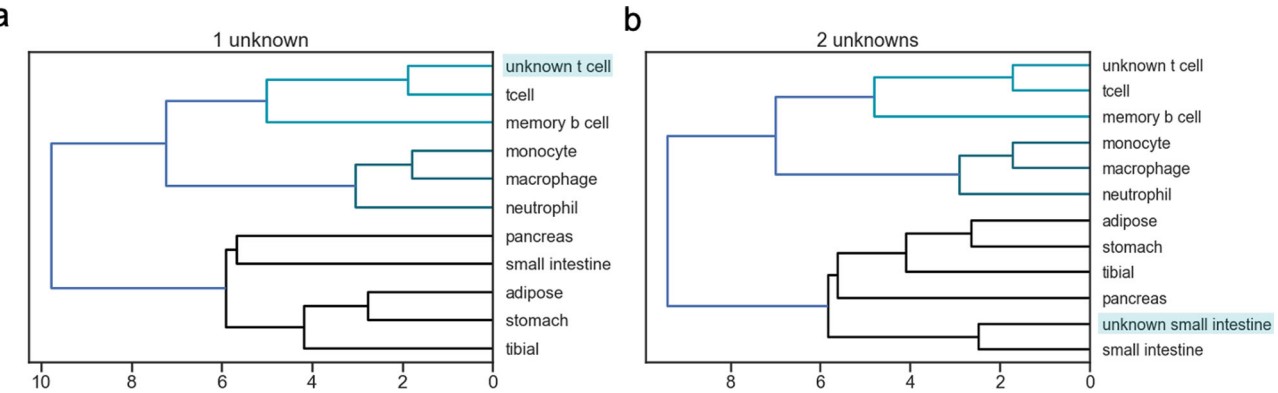

**Fig. 6 Hierarchical clustering of the CelFiE methylation proportion estimates for (a) one unknown and (b) 2 unknowns with the true WGBS methylation proportions.** The shaded blue box indicates the unknown tissue. The light blue, dark blue, and black colors indicate clusters of tissues detected by the hierarchical clustering algorithm.

proportions for CD4+ T cells and the small intestine. To do this, we appended the methylation proportions learned by CelFiE for the two unknown cell types to the matrix of true reference methylation proportions, including the values for T cells and small intestine that were originally masked. We calculated a distance matrix for the reference matrix plus unknowns and used this to perform hierarchical clustering. Figure 6 shows that the unknown cell types were segregated with their true cell type. For the case of one unknown, the unknown that was truly T cell clusters with the reference T cell sample. Furthermore, the average Pearson's correlation between the learned unknown cell type methylation proportions and the reference T cell methylation proportions was higher than all other cell types, $r^2 = 0.95$, suggesting that CelFiE learned the correct cell type for one unknown. For the two unknown cell types, unknown 1 remained clustered with the reference CD4+ T cell sample and had a high correlation with the reference CD4+ T cell methylation patterns, $r^2 = 0.94$. Unknown 2 clustered with the reference small intestine sample along with other gastrointestinal issues. The correlation between the estimated and true small intestine methylation values was the highest of all pairings, $r^2 = 0.87$. Together with the data

presented in Fig. 5b, these observations suggest that even with an incomplete reference, CelFiE estimates both the correct cell type proportion and cell type methylation values.

**Application to pregnancy**. To validate CelFiE, we first choose to analyze cfDNA from pregnant and non-pregnant females since these populations provide a robust example of a verifiable positive and a control group[44]. Unlike the decomposition of cell types in blood, there is no FACS or similar existing standard for cfDNA. Nonetheless, we know a priori that non-pregnant women will not have placenta cfDNA in their bloodstream.

To test CelFiE in pregnant and non-pregnant women, we downloaded publicly available WGBS cfDNA of 7 pregnant and 8 non-pregnant women[45]. All women were between 11 and 25 weeks gestation at the time of cfDNA extraction. Next, we subset the WGBS sites to the same TIMs we use in the previous section and summed all reads ±250 bp around each TIM (see the "Methods" section). Twenty WGBS datasets from the ENCODE and BLUEPRINT projects were chosen for the reference panel, representing tissues and cell types throughout the body and blood, along with one unknown category. The decomposition

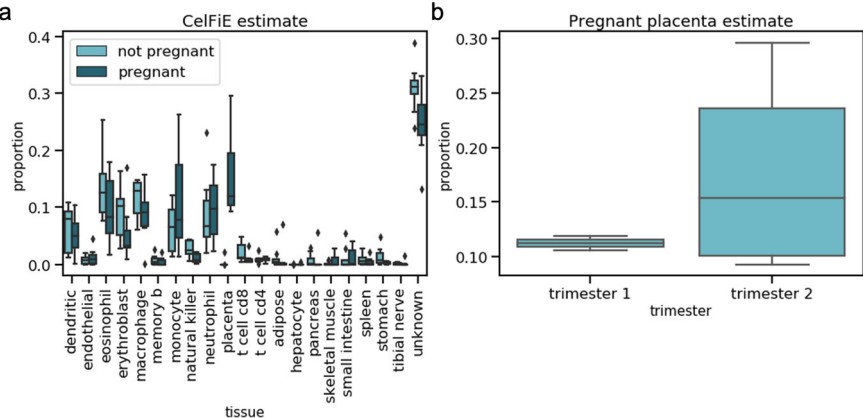

**Fig. 7 Decomposition estimates for cfDNA derived from pregnant women and non-pregnant controls. a** CelFiE decomposition estimates for independent samples of $n = 8$ non-pregnant (light blue) and $n = 7$ pregnant women (dark blue). **b** CelFiE placenta estimates for $n = 3$ pregnant women in the first trimester and $n = 4$ women in the second trimester. In all cases, the center line of the boxplot indicates the mean, the outer edges of the box indicate the upper and lower quartiles, and the whiskers indicate the maxima and minima of the distribution.

result is the random restart with the highest log-likelihood of 10 total restarts.

CelFiE estimated a high proportion of white blood cells (dendritic cells, eosinophils, monocytes, neutrophils, etc.), consistent with previous estimates based on cfDNA and our expectation that blood cells have high rates of cell turnover[25,46]. CelFiE detected a small proportion of cfDNA coming from gastrointestinal tissues, such as the small intestine or stomach, which may also be due to the relatively high cell shedding in these tissues[47]. We used a single unknown cell type component and we estimated that it is large, with a mean of $0.31 \pm 0.04$ in non-pregnant women and a mean of $0.25 \pm 0.06$ in pregnant women (Fig. 7a). To better understand which tissues and cell types are driving the unknown component, we performed hierarchical clustering on the estimated methylation values for the unknown component with the methylation values for the known cell types contained in the reference panel (Fig. S11). We found that it clustered most closely with endothelial cells. This suggests that as reference panels improve, there is additional biological insight that may be gained by using CelFiE.

To evaluate which cell types differ the most between pregnancy states, we performed grouped two-sample $t$-tests of inferred cell type proportions. As expected, placenta showed the greatest difference, ranging from 9.3% to 29.7% (median 11.9%), and $2.9 \times 10^{-16}$ to $2.1 \times 10^{-2}$ (median $2.3 \times 10^{-12}$) in pregnant and non-pregnant women, respectively (two-sided grouped $t$-test, $p = 4.5 \times 10^{-5}$). We also found that CelFiE estimated a higher placental component in the second trimester (median 11.2% in trimester 1 and median 15.3% in trimester 2), concordant with the growth of the placenta throughout pregnancy (Fig. 7b). This is also consistent with previous estimates of the proportion of placental DNA in the cfDNA of pregnant women (median 15.3% in trimester 1/2)[48]. We restricted statistical tests to the relevant tissue, in this case the placenta, but estimates are provided for all tissues and cell types in Fig. 7.

To further validate our method, we compared CelFiE predictions with those from our WGBS adaption of MethAtlas, least-squares regression, and our projection method (Fig. S12A–C). While these methods are not explicitly designed to be ran on WGBS data, all three methods estimated a higher proportion of placental cfDNA in pregnant women than in non-pregnant women, as we expected (Supplementary Table S1). Least-squares regression, however, produced negative estimates, suggesting that this method is unsuitable for real data applications. Furthermore, all three methods estimated proportions of

blood cell types that may be inconsistent with known cell-type proportions in whole blood. For example, all methods estimated a large erythroblast component, on average about 24%. This was higher than expected since nucleated red blood cells are generally rarer than white blood cells in the blood[49]. Furthermore, white blood cells, such as neutrophils, have a much higher turnover rate, making them more likely to appear in cfDNA[50]. While the high proportion of erythroblasts may indicate the presence of a red blood cell precursor not captured by the current reference panel, it may also be a consequence of a bias introduced by missing tissues in the reference panel. For instance, CelFiE ran with an unknown component on the same data estimated an erythroblast proportion of $0.073 \pm 0.052$. When CelFiE was ran without an unknown component (Fig. S12D), the erythroblast proportion increased to $0.29 \pm 0.11$. This could suggest that, as seen in Fig. S5, decomposition estimates without unknown components may cause overestimation of other cell types in the mixture.

**Application to ALS**. Lastly, we examined cfDNA in ALS patients and age-matched controls (see the "Methods" section). ALS patients represented a range of disease severity and onset sites. We first examined the overall abundance of cfDNA in cases ($n = 28$, mean $297.72 \pm 110.57$ pg/ul) and controls ($n = 25$, mean $218.78 \pm 139.17$ pg/ul). We observed a significant excess in cases (Fig. 8a, $p = 5.00 \times 10^{-3}$), but it was unknown what tissue or tissues are responsible for this increase. To explore possible overrepresented tissues in ALS cfDNA, we applied CelFiE first to a discovery cohort composed of 8 controls and 8 cases from both the University of Queensland and UCSF (Fig. S13A). As with the pregnancy cfDNA, we confined the WGBS data to TIM sites and then summed $\pm 250$ bp around the TIMs. We decomposed all mixtures using the same reference tissues as in the pregnancy decomposition, and one unknown. We restricted statistical tests to two biologically relevant tissues for ALS: skeletal muscle and tibial nerve. Notably, we found a difference in the estimated skeletal muscle proportions, specifically finding an excess in cases relative to controls ($p = 5.02 \times 10^{-2}$) (Fig. S14A).

We validated this difference with an independent replication of 8 cases and 8 controls from University of California San Francisco (UCSF) for which WGBS was performed. As expected, we found that the mixture was composed largely of blood cells (Fig. S13B), with the top 5 tissues by proportion being neutrophils, monocytes, macrophages, eosinophils, and erythroblasts. In addition, CelFiE estimated a large unknown component, with a

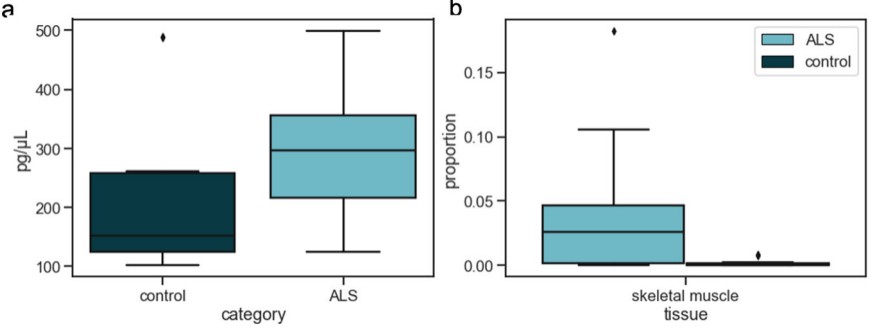

**Fig. 8 CfDNA concentration and decomposition estimates for ALS patients and age-matched controls. a** CfDNA concentrations for $n = 28$ independent cases and $n = 25$ independent controls and **b** CelFiE skeletal muscle estimates for $n = 16$ ALS patients (light blue) and $n = 16$ controls (dark blue) from both UCSF and University of Queensland. In both panels, the center line of the boxplot indicates the mean, the outer edges of the box indicate the upper and lower quartiles, and the whiskers indicate the maxima and minima of the distribution.

mean proportion of $0.42 \pm 0.11$ for ALS cases and $0.30 \pm 0.19$ for control samples. This large unknown component did not cluster closely with any cell type or tissue contained in our reference panel when we applied hierarchical clustering on the CelFiE estimate of the unknown methylation values (Fig. S15), which could indicate that CelFiE captured a substantial signal not captured by other methods. Furthermore, we replicated the significantly higher skeletal muscle component in ALS cases, with a mean muscle proportion of $0.057 \pm 0.06$, while CelFiE estimated an average proportion of $8.9 \times 10^{-4} \pm 1.3 \times 10^{-3}$ in cases (grouped $t$-test $p = 7.8 \times 10^{-3}$) (Fig. S14B). CelFie ran on the combined data (Fig. S13C), estimated a mean proportion of $0.038 \pm 0.020$ in ALS samples and $1.7 \times 10^{-3} \pm 2.6 \times 10^{-3}$ in controls (grouped $t$-test $p = 2.4 \times 10^{-3}$) (Fig. 8b).

Finally, we ran least-squares regression, our projection method, and MethAtlas on our combined ALS cfDNA data (Fig. S16). As in Fig. S12, we found that these methods estimated higher proportions of erythroblasts than CelFiE, and that least-squares regression produced negative estimates. We did find, however, that all three methods recapitulated our finding of a higher proportion of skeletal muscle in ALS patients (Supplementary Table S2). While these differences are similar in magnitude to those from CelFiE, they are less significant (least squares: $p = 0.019$; projection method: $p = 0.026$; MethAtlas: $p = 0.012$), possibly due to the higher error in these methods.

Together, these results suggest that cfDNA is a promising direction to identify the first quantitative biomarker for muscle atrophy and death that is a hallmark of ALS.

## Discussion
During disease or increased cell turnover, elevated levels of cfDNA can be detected in the blood. For example, increases in the amount of cfDNA have been detected in patients with multiple types of cancer, autoimmune diseases, as well as acute episodes of myocardial infarction, trauma, transplantation response, and exercise[51–53]. Correspondingly, the utility of cfDNA as a diagnostic biomarker has been demonstrated in an increasing number of settings, including prenatal testing[54] and the detection of tumor-specific mutations[55,56]. Of great interest, however, is that assessments of cfDNA can now also provide information about cfDNA cellular origin[24–27]. This type of qualitative and quantitative assessment presents an individualized, unbiased approach to understanding cellular turnover over time. However, these technologies are nascent, noisy, and expensive.

In this work, we presented an algorithm, CelFiE, to decompose complex cfDNA mixtures into their cell types of origin. CelFiE can accurately decompose cfDNA mixtures with low sequencing coverage in both the reference cell types and the patient cfDNA samples. We also showed that CelFiE could estimate cell type proportions using relatively few sites, and that its performance improves as more tissue informative sites are selected. This could indicate CelFiE's utility in methylation capture panel development, where highly informative sites are selected and sequenced to high depth[57]. Furthermore, as cohort sizes are expanded, it can accurately estimate multiple unknown cell types, which reduces bias and increases confidence in the decomposition. Finally, the EM algorithm underlying CelFiE is computationally efficient, with iteration cost scaling linearly with the number of samples, CpG sites, and cell types.

We began by validating CelFiE extensively in simulations. In the context of simulated low read-count methylation data, CelFiE outperformed linear least-squares regression, our L1-projection method, and MethAtlas, another cfDNA decomposition method. Since these methods are not explicitly designed for this data regime, CelFiE's improvements may make it a useful addition to the tools available to cfDNA researchers. To further demonstrate the accuracy of CelFiE, we applied it to real data from pregnant women. Decomposition estimates of placenta from pregnant women were significantly different from non-pregnant women. This provided a natural validation for CelFie, illustrating that it can correctly learn differences in cfDNA cell type of origin, even in real data sets.

In our study of ALS patients, we found that cfDNA levels are increased in ALS cases compared to controls. To understand what cell types are driving this difference, we applied CelFiE to the cfDNA samples, finding significantly higher skeletal muscle in patients with ALS. Future work will expand on this result by expanding the cohort size, and by testing for associations between cell type of origin and disease progression or severity. We may also test for associations between decomposition estimates and disease onset site. Furthermore, as cohort sizes expand, we will have the power to estimate multiple unknown categories. These multiple unknown categories could be used to further subtype ALS cases. We consider the current results, a promising step forward, especially as ALS currently has no reliable biomarker. These results also suggest that CelFiE might prove useful for quantifying cell death in other complex diseases.

The accuracy of CelFiE depends on several factors including read depth, the cell-type specificity of the sites considered, the abundance of key cell types, and the quantity and quality of reference data and cfDNA patient samples. Recent technologies for digesting or capturing specific regions of cfDNA[58], may allow deeper sequencing of informative CpGs. Selecting such TIM CpGs demonstrated marked improvement in accuracy and could be used to select sites for capture.

There are a number of areas for improvement. Many of the reference samples used here were complex mixtures of cell types and could be modeled as such, similar to the recent approach, FEAST[59], which modeled reference mixtures of microbial communities. Moreover, WGBS simulation results showed a high degree of correlation between replicates, but we believe modeling inter-person heterogeneity will likely improve the results further in real cfDNA samples. We currently account for the local correlation of CpG methylation by summing proximal CpG methylation states, but nearby CpGs may not always convey identical cell type information. Future work could also focus on modeling the relationship between cell types and tissues. For example, since cell types are correlated in their methylation profiles, it could be interesting to consider a hierarchical model in which the composition can be considered at different levels of cell type phylogeny[60]. This may help us gain additional power to identify samples, particularly highly similar cell types or tissues. Finally, the addition of non-CpG methylation and cfDNA fragment length may provide additional sources of information about cell types of origin.

In summary, we present CelFiE, an efficient EM algorithm for decomposing cfDNA mixtures into their cell type of origin, even when the data are low count or noisy. CelFiE can additionally robustly estimate both known and unknown cell types in cfDNA. Overall, our work demonstrates that CelFiE could be a useful tool for quantifying cell death, applicable to biomarker discovery and disease monitoring.

## Methods

**CelFiE overview.** We assume that we are provided with a bisulfite sequenced reference data set, composed of $T$ cell types indexed by $t$, at $M$ CpG sites indexed by $m$. Bisulfite sequencing produces read counts from specific cell types that we collect in two $T \times M$ matrices: $Y$ and $D^Y$, where $Y_{tm}$ and $D^Y_{tm}$ are the number of methylated and total reads at CpG $m$, respectively, in reference cell type $t$. Together, these two matrices represent the reference cfDNA data.

We are also provided with cfDNA extracted from $N$ individuals indexed by $n$. The bisulfite sequencing read counts of the cfDNA are given in two $N \times M$ matrices $X$ and $D^X$, with $X_{nm}$ and $D^X_{nm}$ giving the number of methylated and total reads at CpG $m$ in the cfDNA from individual $n$, respectively. These two matrices represent the sample cfDNA data.

CelFiE takes as input the matrices $Y$, $D^Y$, $X_{nm}$, and $D^X_{nm}$, and then outputs a matrix $\alpha$, where $\alpha_{nt}$ is the fraction of the cfDNA in person $n$ that originated from cell type $t$.

**Model.** We model the cfDNA as a mixture of DNA from cell types in the reference panel and, potentially, unknown cell types absent from the reference panel. We assume that the individuals are independent given the true, unknown methylation proportions of each cell type, and the individual-specific cell type proportions.

We assume that reference data are drawn from a binomial distribution:

$$Y_{tm} | D^Y_{tm}, \beta_{tm} \overset{iid}{\sim} \text{Binomial}\left(D^Y_{tm}, \beta_{tm}\right) \tag{1}$$

where $\beta_{tm} \in [0, 1]$ is the true, unknown proportion of DNA in a cell type that is methylated at position $m$. This model assumes no intra-cell type heterogeneity, in the sense that each cell in a cell type has identical methylation probability.

Next, we model the samples in the cfDNA data. We assume each cfDNA read is drawn from some cell type $t$ at some marker $m$, and in turn that its methylation value is drawn from a Bernoulli distribution governed solely by the methylation proportion in the cell type of origin:

$$x_{nmc} | \beta, Z_{nmc} = t \overset{iid}{\sim} \text{Bernoulli}\left(\beta_{tm}\right) \tag{2}$$

where $x_{nmc}$ is the methylation status of the $c$-th read from sample $n$ at position $m$, and $Z_{nmc} = t$ indicates that it is the cell type of origin for this read. For each person and methylation site, we define the total number of methylated reads as $X_{nm} := \sum_{c=1}^{D^X_{nm}} x_{nmc}$. This simply sums the methylation status over all reads for each person at each site. In the special case where $D^X_{nm} = 0$, we define $X_{nm} = 0$.

Finally, we assume that the cell type of origin of each cfDNA molecule is drawn independently from some individual-specific multinomial distribution:

$$Z_{nmc} | \alpha_n \overset{iid}{\sim} \text{Multinomial}\left(\alpha_{n1}, \dots, \alpha_{nT}\right) \tag{3}$$

where $\alpha_{nt}$ is the probability that a read from person $n$ comes from cell type $t$.

**EM algorithm for one cfDNA sample.** For simplicity, we first describe CelFiE in the case where the cfDNA data set contains only a single person, meaning the decomposition relies almost exclusively on the reference panel. We then explain how CelFiE can jointly model multiple individuals in the cfDNA data, as well as how and why this enables the estimation of unknown cell types. Full details of both algorithm derivations are given in the Supplement.

Formally, assume there is only one sample in the cfDNA data (i.e. $N = 1$). We define $z_{tmc}$ as a binary indicator for whether read $c$ at CpG $m$ for the single cfDNA individual originates from cell type $t$. In relation to $Z$ above, $z_{tmc} = 1$ if $Z_{1mc} = t$, and otherwise 0. That is, $Z_{1mc}$ is a categorical variable, and $z_{tmc}$ indicates which value $Z_{1mc}$ takes.

To calculate the full data likelihood, $P(x, z, Y | \alpha, \beta)$, we first factorize it into $P(x, Y | z, \alpha, \beta) \cdot P(z | \alpha, \beta)$. This then simplifies into three components:

$$P(x, z, Y | \alpha, \beta) = P(x | z, \beta) P(z | \alpha) P(Y | \beta) \tag{4}$$

The first component defines the probability of the cfDNA reads, given which cell type they come from and the methylation proportions of those cell types. The third component analogously defines the probability of drawing the reference reads. The second component describes the probability of observing a specific cell type in the cfDNA, which is determined by the proportion of each cell type in the individual's cfDNA.

We show in the supplement that the resulting log-likelihood is equivalent to:

$$\sum_{t,m,c} z_{tmc} \left[ x_{mc} \log\left(\beta_{tm}\right) + (1 - x_{mc}) \log\left(1 - \beta_{tm}\right) \right] + \sum_{t,m,c} z_{tmc} \log \alpha_t$$
$$+ \sum_{t,m} \left( Y_{tm} \log \beta_{tm} + (D^Y_{tm} - Y_{tm}) \log(1 - \beta_{tm}) \right) \tag{5}$$

For this one-sample section, we drop an index on $x$ and write $x_{mc}$ instead of $x_{1mc}$. Analogously, we write $X_{nm} = X_m$ as the total number of methylated reads at position $m$ (and $D^X_{nm}$ as $D^X_m$).

To calculate the expected log-likelihood, i.e., the $Q$ function, we must integrate over the conditional distribution for the missing data, i.e. $P(z | x, \beta, \alpha)$. Since $z_{tmc}$ is binary and each read and site is assumed independent, this distribution is the probability that each $z_{tmc}$ is 1. In other words, the probabilities that each read comes from each cell type are sufficient statistics, and are given by

$$P(z_{tmc} = 1 | x_{mc}, \beta, \alpha) = \frac{\beta_{tm}^{x_{mc}} (1 - \beta_{tm})^{1 - x_{mc}} \alpha_t}{\sum_k \beta_{kt}^{x_{mc}} (1 - \beta_{kt})^{1 - x_{mc}} \alpha_k} =: \tilde{p}_{tmc}(\alpha, \beta) \tag{6}$$

Conceptually, if read $c$ is methylated, this indicates the read is more likely to come from cell types with high methylation proportion, as $\beta_{tm}$ is larger (and vice versa if the read is unmethylated). Regardless the methylation state, however, this equation also says that the read is likelier to come from more common cell types, as $\alpha_t$ is larger.

This final term $\tilde{p}_{tmc}(\alpha, \beta)$, seems complex. However, it actually only depends on the specific read $c$ through its methylation status, and takes only two values. We can redefine it in simpler terms, which represents the probability of each cell type for each read depending on its methylation:

$$\begin{aligned}\frac{\beta_{tm} \alpha_t}{\sum_k \beta_{kt} \alpha_k} &=: p_{tm1}(\alpha, \beta) = \tilde{p}_{tmc}(\alpha, \beta) \text{ if } x_{mc} = 1 \\ \frac{(1 - \beta_{tm}) \alpha_t}{\sum_i (1 - \beta_{kt}) \alpha_k} &=: p_{tm0}(\alpha, \beta) = \tilde{p}_{tmc}(\alpha, \beta) \text{ if } x_{mc} = 0\end{aligned} \tag{7}$$

*E step.* The $Q$ function is defined at iteration $i$ by

$$Q_i(\beta, \alpha) := \mathbb{E}_{z | x, \alpha^{(i)}, \beta^{(i)}}(\log P(x, z, y | \alpha, \beta)) \tag{8}$$

where $\alpha^{(i)}$ and $\beta^{(i)}$ are the parameter estimates of the cell type proportions and methylation proportions from the last EM step. Let $p^{(i)}_{tm} := p_{tm1}(\alpha^{(i)}, \beta^{(i)})$, which is the probability that a methylated read at site $m$ comes from cell type $t$ given the previously estimated parameters from iteration $i$. Then $Q_i$ is

$$Q_i(\beta, \alpha) = \sum_{t,m} \left[ \left( Y_{tm} + p^{(i)}_{tm1} X_m \right) \log\left(\beta_{tm}\right) + \left( D^Y_{tm} - Y_{tm} + p^{(i)}_{tm0}(D^X_m - X_m) \right) \log\left(1 - \beta_{tm}\right) \right]$$
$$+ \sum_{t,m} \left( X_m p^{(i)}_{tm1} + (D^X_m - X_m) p^{(i)}_{tm0} \right) \log \alpha_t \tag{9}$$

The first line in this equation captures the expected total number of methylated reads (first term in the sum) and the total number of expected unmethylated reads (second term) for each cell type and site. Each of these terms combines both the reference and cfDNA contribution, e.g. the first term combines the total methylated reads from the relevant reference cell type ($Y_{tm}$) with the expected number of methylated reads from that cell type in the cfDNA mixture ($p^{(i)}_{tm1} X_m$).

Complementary to the first line, the second line determines the likelihood of $\alpha$ and does not depend on $\beta$. It captures the likelihood of observing the expected cell type frequencies. This is given by the sum of the expected methylated and expected unmethylated reads over all loci.

*M step.* To update the estimated cell type proportions, $\alpha$, we maximize $Q_i$ under the constraint that $\alpha$ is a probability vector, i.e., its entries are non-negative and sum to

one. The maximizer is

$$\alpha_t = \frac{\sum_m \left( x_m p_{tm1}^{(i)} + (D_m^X - x_m) p_{tm0}^{(i)} \right)}{\sum_{k,m} \left( x_m p_{km1}^{(i)} + (D_m^X - x_m) p_{km0}^{(i)} \right)} \qquad (10)$$

The numerator is simply the number of reads expected to originate from each cell type, which is calculated by adding the expected contributions from the methylated and the unmethylated reads. The proportions are then obtained by normalizing these numerators to sum to 1.

The other M step update is for $\beta$, the proportion of reads that are methylated at each site and in each cell type:

$$\beta_{tm} = \frac{p_{tm1}^{(i)} x_m + Y_{tm}}{p_{tm0}^{(i)}(D_m^X - x_m) + D_{tm}^Y - Y_{tm} + p_{tm1}^{(i)} x_m + Y_{tm}} \qquad (11)$$

Intuitively, this is the ratio of the expected number of methylated vs. total reads from cell type $t$ at site $m$. This update is conceptually similar to the $\alpha$ update in the sense that it matches an estimated proportion to an expected proportion. For $\alpha_t$, this is the expected proportion of reads deriving from cell type $t$; for $\beta_{tm}$, this is the expected proportion of reads from cell type $t$ that are methylated at site $m$.

**EM algorithm for multiple cfDNA samples.** We now return to allowing $N > 1$ cfDNA samples. In this setting, $\alpha$ is a matrix, because each cfDNA sample may have different proportions of each cell type in their cfDNA mixture. Further, $x_{nmc}$ and $Z_{nmc}$ are now 3-dimensional arrays indexed by cfDNA individual $n$, methylation site $m$, and sequencing read $c$, and the binary indicators $z_{nmtc}$ are now 4-dimensional, as they additionally index each cell type.

The conditional distribution for $z$ at each step of the EM algorithm now becomes:

$$P(z_{ntmc} = 1 | x_{nmc}, \beta, \alpha) = \frac{\beta_{tm}^{x_{nmc}}(1 - \beta_{tm})^{1 - x_{nmc}} \alpha_t}{\sum_k \beta_{tk}^{x_{nmc}}(1 - \beta_{tk})^{1 - x_{nmc}} \alpha_k} =: \tilde{p}_{ntmc}(\alpha_n, \beta) \qquad (12)$$

As before, this $\tilde{p}$ term depends on $c$ only through $x_{nmc}$, and so we simplify terms by defining $\tilde{p}_{ntmc}(\alpha_n, \beta) = p_{ntmj}(\alpha_n, \beta)$ if $x_{nmc} = j$ for $j = 0, 1$.

To simplify the E step, we define the responsibilities by $p_{ntmj}^{(i)} := p_{ntmj}(\alpha_n^{(i)}, \beta^{(i)})$. For $j = 0$, this gives the conditional probability that an unmethylated read from individual $n$ at site $m$ comes from cell type $t$ given the current parameter estimates; $j = 1$ gives the analogous probability for methylated reads. Since we assume cfDNA individuals are independent given $\alpha$ and $\beta$, the E step is a simple generalization of the one-sample E step that sums over samples and can be written:

$$Q_i(\alpha, \beta) = \sum_{n,t,m} \left[ \left( Y_{tm} + p_{ntm1}^{(i)} X_{nm} \right) \log(\beta_{tm}) + \left( D_{tm}^Y - Y_{tm} + p_{ntm0}^{(i)}(D_{nm}^X - X_{nm}) \right) \log(1 - \beta_{tm}) \right]$$

$$+ \sum_{n,t,m} \left( X_{nm} p_{ntm1}^{(i)} + (D_{nm}^X - X_{nm}) p_{ntm0}^{(i)} \right) \log \alpha_{nt} \qquad (13)$$

This $Q$ function can be interpreted identically to the single-sample $Q$ function. The only difference is that now reference reads are added with expected cfDNA reads for multiple individuals, and the expectations $(p_{ntmj}^{(i)})$ depend on cfDNA individual $n$ as well as cell type $t$, CpG site $m$, and methylation status $j$.

$Q_i$ additively splits over row of $\alpha$, therefore, the updates for each $\alpha_n$, are identical to the single-sample $\alpha$ updates, where $\alpha_{nt}$ replaces $\alpha_t$, $X_{nm}$ replaces $X_m$, $D_{nm}^X$ replaces $D_m^X$, and $p_{ntmj}^{(i)}$ replaces $p_{tmj}^{(i)}$. This means that if we condition on the number of reads coming from each cell type in person $n$, the estimates of that person's cell type proportion do not depend on anything else.

For $\beta_{tm}$, the M-step again compares the expected number of methylated and unmethylated reads at CpG $m$ from cell type $t$, where the expectation combines reads from reference cell type $t$ with the expected number of cfDNA reads from cell type $t$. The only difference is that now the expectation combines the expected contributions from multiple cfDNA samples:

$$\beta_{tm} = \frac{\sum_n p_{ntm1}^{(i)} X_{nm} + Y_{tm}}{\sum_n p_{ntm0}^{(i)}(D_{nm}^X - X_{nm}) + D_{tm}^Y - Y_{tm} + \sum_n p_{ntm1}^{(i)} X_{nm} + Y_{tm}} \qquad (14)$$

*Unknown sources.* It is likely that there are cell types in the cfDNA mixture not contained in the reference data. To estimate the proportion of an unknown cell type with CelFiE, we append a zero row to $D^Y$ and $Y$, and then run CelFiE as usual. This produces an EM that is mathematically similar to the STRUCTURE model of mixtures of human populations[61]. Essentially, CelFiE estimates methylation patterns and abundances for the unknown cell type(s) that maximize the overall likelihood. To model more than one unknown cell types, additional rows of zeros are added to $D^Y$ and $Y$. Note that if the number of unknown cell types is greater than the number of individuals, the problem is not identified.

*Regularization and missing data.* Missing observations are allowed in both the reference and the input. It is represented as a 0 entered in both $X/D^X$ or $Y/D^Y$. In practice, we add a methylated and unmethylated pseudocount to every entry of $X$ and $Y/D^X$ and $D^Y$ to stabilize the algorithm and likelihood in case of cell type/site combinations with very low coverage.

*Computational cost.* Each iteration of the EM algorithm in CelFiE involves three calculations. First, $p_{ntmj}^{(i)}$ is evaluated for each sample $n$, cell type $t$, CpG site $m$, and methylation status $j = 0, 1$; each calculation is independent of the input data dimensions, hence evaluating $p^{(i)}$ is $O(NTM)$. Second, $\alpha_{nt}$ must be evaluated, which involves summing over $M$ sites for each $n$ and $t$, giving overall complexity $O(NTM)$. Finally, updating $\beta_{tm}$ requires summing over all cfDNA individuals and the reference cell type data, again giving overall complexity $O(NTM)$. Overall, this means that CelFiE scales linearly in sample size, number of CpGs, and number of cell types.

We also note that if multiple references were included, the cost would not multiply–rather, the cost would increase to $O((N + N_{ref})TM)$, where $N_{ref}$ is the (maximum) number of reference samples per cell type.

**Other decomposition methods.** Linear least-squares regression was implemented using the linregress package from SciPy (v 1.5.2) in Python[62]. We minimized $\min || X\alpha - Y ||_2^2$ where $X$ was the methylation proportions of the cfDNA input and $Y$ was the methylation proportions of the reference matrix. We estimated $\alpha$, which was the cell-type proportions of the cfDNA mixture. Since least-squares regression does not return estimates that sum to one, we divided $\alpha$ by its sum.

Projection onto the L1 ball was a implemented in a custom Python script available at https://github.com/christacaggiano/celfie. There, we optimize a binomial log-likelihood, where the number of successes is the number of methylated cfDNA reads, the number of trials is the cfDNA read depths, and the probability of success is the reference methylation values multiplied by the estimate of cell type proportions for a given iteration. Maximum-likelihood optimization was performed using the L-BFGS algorithm in the SciPy Minimize package.

MethAtlas was run using code available at https://github.com/nloyfer/meth_atlas commit #0223493. It was run using the following command: deconvolve.py -a<reference path><ouput directory><samples path>.

**ALS subjects.** ALS patients were recruited jointly from the University of California San Francisco ALS Center and the University of Queensland ALS clinics under clinician supervision. All participants provided informed consent and the study was approved both by the Human Research Ethics Committee at the University of Queensland (IRB 2018002470) and by the UCSF Committee on Human Research (IRB 10-05027).

12 cases and 12 controls from San Francisco and 4 cases and 4 controls from Queensland were included in this study. Controls were from non-related family members or caregivers. cfDNA was extracted after subjects were at rest for more than 30 min to prevent possible confounding from exercise. We collected 20 mL of whole blood from controls and 10 mL from cases, to allow for further analyses.

**ALS cfDNA sequencing.** Whole blood was collected in PAXgene Blood ccfDNA tubes (Qiagen, Cat. No. 768115) and centrifuged at 1900×g for 10 min at RT to isolate plasma. Plasma was centrifuged twice at 16,000×g for 10 min and stored at −80 °C until cfDNA extraction. Circulating cfDNA was extracted from 4 ml (ALS patients) or 8 ml (controls) of plasma using the QIAamp Circulating Nucleic Acid kit (Qiagen, Cat. No. 55114). Larger volumes of control blood were collected to ensure equal amounts of total cfDNA (compared to patients) were analyzed. cfDNA quality and concentration were assessed with an Agilent 2100 Bioanalyzer, using the Agilent High Sensitivity DNA kit (Agilent, Cat. No. 5067-4626). 10 ng of cfDNA were bisulfite-treated and purified using the EZ DNA Methylation-Direct Kit (Zymo Research Cat. No. D5020). Libraries for whole genome bisulfite-sequencing were generated using Accel-NGS® Methyl-Seq DNA Library Kit (Swift Biosciences, Cat. No. 30024) and Accel-NGS Methyl-Seq Dual Indexing kit (Swift Biosciences, Cat. No. 38096), with eight cycles of indexing PCR. Libraries were quantified by qPCR with the Hyper Library Quantification kit (Kapa, Cat. No. KR0405) and paired-end sequenced on a NovaSeq 6000 System (Illumina).

**ALS cfDNA data processing.** Our ALS case-control WGBS data (including both the UCSF and UQ data) were processed according to the ENCODE consortium guidelines[37]. Quality of the fastq files was assessed using FastQC (v 0.11.9)[63]. All samples had average phred scores ≥28. Adapters were trimmed from the paired end fastq files using TrimGalore (v 0.6.6). Four basepairs were trimmed from the 5′ direction and 12 base pairs were trimmed from the 3′ direction. Trimmed fastq files were mapped to a bisulfite converted hg38 genome using the Bismark (v 0.23.0) implementation of Bowtie2 (v 2.3.5.1). CpG methylation was from a Samtools (v 1.7) sorted Bismark generated bam file using MethylDackel (v 0.5.0). For this study we were only interested in CpG methylation, which is largely symmetric. Thus, we combined reads on each strand, using the MethylDackel "–mergeContext." option. To standardize methylation calls across all WGBS data sources, hg38 coordinates were reported as 0-indexed. All packages were installed using Anaconda (v 4.9.2). For more details, see https://github.com/christacaggiano/ENCODE_WGBS.

**Pregnancy cfDNA data processing.** Data from pregnant women and non-pregnant controls were taken from Jensen et al. at Raw fastq files from were retrieved from dbGaP identifier phs000846. To ensure consistency across cfDNA samples, data was processed identically to 1. In the original Jensen et al. study

design, multiple fastq files mapped to one sample. Thus, after methylation calling, we combined the appropriate methylation bed files into one per individual, for a total of 15 bed files.

**WGBS simulation data**. Ten adult (small intestine, pancreas, monocyte, stomach, tibial nerve, macrophage, memory B cell, adipose, neutrophil, and T cell) WGBS bedMethyl files were obtained from the ENCODE and BLUEPRINT project[37,39] (Data identifiers described in Supplementary Data 1). BLUEPRINT data was downloaded as two bigWig files, a methylation signal bigWig and a coverage of methylation signal bigWig. These files were combined into one bedgraph-format file using the UCSC bigWigToBedGraph utility.

Each WGBS file had two biological replicates coming from distinct people. All bed file coordinates were harmonized to hg38 using hgLiftOver[64]. For each tissue or cell type, the file was restructured to report the number of methylated reads and read depth for each CpG locus. Coordinates were standardized to be zero-indexed.

**WGBS reference data**. Reference data for the real cfDNA decomposition experiments in 1 and 1 were retrieved from ENCODE and BLUEPRINT. Twenty tissues and cell types were chosen to be representative of the many tissues possible in cfDNA. To decrease noise, we combined two replicates of the tissue when available (see Supplementary Data 1 for individual accession numbers). As described previously, we mapped all data to hg38, and converted the coordinates to be 0-indexed.

**Site selection and summing**

*Tissue informative markers*. Only about 20% of autosomal CpGs vary by cell type[41]. Selecting sites that do vary enriches for information on tissue of origin and reduces the EM computational burden, which scales linearly in the number of sites. We propose selecting tissue informative markers (TIMs) without curation, an approach inspired by ancestry informative markers in population genetics[65,66].

After processing the WGBS files, one replicate per tissue was segregated into a reference matrix. This reference matrix was used to calculate TIMs. We assess whether a CpG is a TIM one locus at a time. For each CpG, the distance between the percent methylation of that cell type and the median percent methylation for that CpG was calculated. Only CpGs where the median depth was greater than 15 and had no missing data were considered. The top $N$ (default = 100) CpGs with the greatest distance per cell type were selected. TIMs provide increased accuracy in decomposition, and vastly improve computation time. We reference cell types to have overlapping TIMs (i.e., one CpG may be a TIM for both pancreas and liver). We combine proximal CpGs (±250bp) around TIMs to increase confidence in the methylation state for a particular CpG (see the section "Site combination"). To test the performance of TIMs, we create a complex mixture of 10 WGBS samples and calculate 100 TIMs per sample (for a total of 1000 CpGS). We compared CelFiE decomposition estimates using 1000 random summed 500bp regions, 1080 500bp regions published in Sun et al.[48], and our TIM regions. For the data set of WGBS mixtures, TIMs perform better than random and better than the Sun et al. regions (Fig. S8). We believe that TIMs will be especially desirable for downstream applications, where permuting random WGBS CpG sites is not feasible, or in the development of a capture panel (see the "Discussion" section).

*Site combination*. To demonstrate whether summing sites improves CelFiE's ability to discriminate tissues, we create complex mixtures of WGBS samples, as in the previous section. We either use single TIMs, or add all methylated and unmethylated counts for all CpGs ±250bp around a TIM. Summing CpGs ±250 improves the performance of CelFiE (Fig. S9).

**Reporting summary**. Further information on research design is available in the Nature Research Reporting Summary linked to this article.

## Data availability

Raw and processed WGBS data generated for this study are available at NCBI GEO under accession number GSE164600. Tissue and cell-type WGBS data is freely available on the https://www.encodeproject.org/search/?type=Experiment&status=released&perturbed=false ENCODE Project and http://dcc.blueprint-epigenome.eu/#/files BLUEPRINT Epigenome Project data access portals. The pregnancy cfDNA data from Jensen et al[45] used in this study was obtained from dbGaP under accession number phs000846.

## Code availability

Software for CelFiE and the projection method are available at https://github.com/christacaggiano/celfie[67].

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

## Acknowledgements

We thank A. Boudaie for providing software engineering expertise, J. Villagracia for the CelFiE name, and N. Wray for helpful comments on this manuscript. We gratefully acknowledge the funding support provided by the ALS Association, ALS Finding a Cure, the ALS Biomarker Collaboration, the UCSF Weill Award, the Australian National Health and Medical Research Council (NHMRC) and the Motor Neurone Disease Research Institute Australia (MNDRIA). C.C. is supported by T32NS048004. N.Z., J.M., A.D. and C.C. are supported by NIH K25HL121295, U01HG009080, R01HG006399, R01CA227237, R03DE025665, R01ES029929, DoD W81XWH-16-2-0018, ALS Association, ALS Finding a Cure, the ALS Biomarker Collaboration, and the UCSF Weill Award. F.C.G is supported by the NHMRC (1078901, 1113400, 1121962), a MNDRIA grant-in-aid (2017) and F.C.G a MNDRIA Susie Harris Travel Fellowship. B.L.B. and B.C. were supported by grants from the ALS Association (ALSA #17-IIP-358) and the NIH (HL064658, HL146366).

## Author contributions

C.C., A.D., J.M., and N.Z. conceived and designed the CelFiE statistical model. C.C., A.D., and N.Z. designed the computational experiments. C.C. wrote all code and carried out computational experiments. C.C. and F.G. planned and carried out the next-generation sequencing bioinformatics. B.C., B.L.B., and N.Z. conceived the ALS application and study design. B.C., B.L.B, and F.G. designed the molecular experiments. B.C. and F.G carried out the molecular experiments. B.C., F.G., C.L.-H., and R.H. were involved in sample collection. C.L.-H. and R.H. provided clinical supervision. C.C. wrote the manuscript with A.D. and N.Z. All authors read, edited, and approved the final manuscript.

## Competing interests

N.Z., B.L.B., B.C., and F.G. have a founding interest in Mercury Epigenomics, LLC, which was not directly or indirectly involved in any of these studies. All other authors report no competing interests.
