## [Peer Review File · Nature Communications]

Reviewers' Comments:

Reviewer #1:

Remarks to the Author:

Overall thoughts:

Overall this paper addresses important challenges in cell type deconvolution and biomarker discovery from cfDNA. Methods like these are especially relevant as the quantity, quality, and diversity of reference data sets continues to grow, allowing this deconvolution approach to potentially provide enough accuracy as to yield important conclusions and diagnoses.

I think what this paper does well is that it provides several relevant motivating examples from real data (ALS, pregnancy) that demonstrate why correct deconvolution can be such a powerful tool. I also believe that it addresses and tackles two particularly hard parts of deconvolution: unknown cell type estimation and rare cell type estimation. These are not easy to get right with more traditional deconvolution methods.

On the other hand, the paper at this point doesn't fully convince the reader as to the advantages of their method over other similar deconvolution methods. There is only one comparison to MethAtlas highlighted in the main text, and this is done on simulated data (although there are a few supplemental figures). More importantly, there is no discussion of what methodological differences there are between the two deconvolution methods (CelFiE vs. MethAtlas) and what aspects of each method may lead to their differing performance. Also, while there may be no other papers that are quite comparable to the CelFiE WGBS application, there are certainly many other papers on cell type deconvolution (for example, deconvolution from RNA-seq data, etc.) that from a methods perspective would make for a very interesting comparison. Have the authors explored other deconvolution methods (least squares regression, non-negative matrix factorization, support vector regression, etc.), and why did they settle on EM?

Specific comments:

1. In simulated data, there appears to be quite a drastic difference between the performance of MethAtlas and CelFiE. It would be useful to describe the methodological improvements made in CelFiE vs. MethAtlas that could attribute to this difference. I also worry that these differences could be related to how the data was simulated instead of true methodological improvements; how can you convince the reader otherwise?
2. There is a lot of focus on simulated data sets, but I think it'd be much more interesting and a better proof of concept to lead with true data applications. I found the real data examples to be much more convincing as for why one should use the CelFiE method.
3. When varying the number of CpG sites in the simulated data sets, were these CpG sites randomly chosen or were only the top 100, 1000, etc. most informative/variable sites chosen? I imagine specifically picking more informative sites will increase performance especially when using only a small number (100) of sites. In general, selecting appropriate CpG sites can have a large impact on deconvolution accuracy. It would be valuable to perform some type of sensitivity analysis showing how for example choosing different numbers of highly informative CpG sites affects the deconvolution results.

Reviewer #2:

Remarks to the Author:

The manuscript by Caggiano et al. entitled "CelFiE: Comprehensive cell type decomposition of

circulating cell-free DNA" develop a method to use WGBS data for low coverage data in cell free DNA in blood named CelFiE (CELI Free DNA Estimation via expectation-maximization). Then they applied this method to investigate cell-free data in pregnant vs non-pregnant women and patients with amyotrophic lateral sclerosis (ALS) vs. age-matched controls. The goal of this work is to overcome several limitations posed by the limited information in microarrays and the small amounts of DNA that are required to be captured in cell-free DNA analyses. An important feature that they underline is how the model is resistant to heavy-tailed distributions and potential model-assumptions. The assumptions in the model and the flexibility of adding unknowns increase the potential applications for this methodology. Said that, I would like some clarifications about your results:

Comment 1: pages 2-3 section 2.2: Although I understand your idea to replicate and compare to Meth_atlas, the sentence is not precise. There are no 25 cell types, instead Moss et al. use both isolated cell-types and tissues, most of them from cancer patients. You may want to adapt your sentence to be more precise. This is mentioned later on section 2.3

Comment 2: page 4 paragraph 3, fig 3: You mention how the accuracy of your results are based on the coverage. What is the minimum coverage for accomplishing a 0.1%? Is it the 1000x, is it lower? Could you, theoretically, capture proportions lower than 0.1% with deeper coverage? When looking at your figure 3 there are differences versus the background and there are significant differences between the true fraction and the detected fraction. However, the median detected values are over the red line with a wide variability. Could you provide some information for the precision of your estimates?

Comment 3: on page 2 section 2.1 you mention the problem of missing cells in your reference. Then you explore one and 2 missing values (Fig S1 and Fig S2) What could be a potential problem for those missing values, if you miss one of the main cells (instead of one at random) could you quantify the bias of your results? Have you quantified the amount of expected bias in both extremes with low proportions and high proportions? You show in the figures some outliers that had correlation below 0.5 and even close to zero. Further, it is not shown the specific tissue/cell-type in your comparison between your method and MethAtlas. Were these results different in specific tissues or were similar independent of the tissue? Please expand on those results.

Comment 4: page 7: in addition to the previous comment you mention a high correlation of the true proportions with you method around the mean of the estimation. I do not see any estimates of the precision of your estimates? How precise were those estimations? How the estimate precision was affected with an increased amount of unknown cells in Fig. 5.

Comment 5: Fig 7: Could you add some information about the unknown fraction observed (>20%) in both pregnant and non-pregnant women?

Comment 6: Fig S8: could you expand about the other components in your deconvolution and their biological meaning? Dendritic cells, small intestines, etc. Are those target cells/organs for the disease? are those random findings?

Comment 7: How is your model handling very similar cell/tissue types? You mention something about small intestine and other GI tissues, are these accurate predictions? are general approximations? Could you add some information about how to contextualize those findings?

We thank the reviewers for their thoughtful remarks and were happy to see that they believe CelFiE makes strides in addressing important issues in cell type decomposition. We have carefully gone through each of the comments and made the appropriate changes. Overall, we believe we have produced a much-improved manuscript. Reviewer comments are in plain text below with our responses in bold.

Reviewer #1 (Remarks to the Author):

Overall thoughts:

Overall this paper addresses important challenges in cell type deconvolution and biomarker discovery from cfDNA. Methods like these are especially relevant as the quantity, quality, and diversity of reference data sets continues to grow, allowing this deconvolution approach to potentially provide enough accuracy as to yield important conclusions and diagnoses.

I think what this paper does well is that it provides several relevant motivating examples from real data (ALS, pregnancy) that demonstrate why correct deconvolution can be such a powerful tool. I also believe that it addresses and tackles two particularly hard parts of deconvolution: unknown cell type estimation and rare cell type estimation. These are not easy to get right with more traditional deconvolution methods.

We thank the reviewer for their positive feedback. We are glad that they found our real-world examples compelling.

On the other hand, the paper at this point doesn't fully convince the reader as to the advantages of their method over other similar deconvolution methods. There is only one comparison to MethAtlas highlighted in the main text, and this is done on simulated data (although there are a few supplemental figures).

We believe a strength of our work is that, to the best of our knowledge, there is no other software designed to be run directly on WGBS cfDNA data. MethAtlas is the closest existing tool but was designed for methylation arrays, and thus requires nontrivial modification to use on our problem. Specifically, to run MethAtlas on our data, we needed to make many bioinformatics decisions on how best to apply it to WGBS; in particular, we found that results were highly sensitive to our choices on transforming WGBS reads into methylation percentages and site selection.

With this caveat, however, we did make improvements to our work to further emphasize the unique advantages of CelFiE as a decomposition method. We applied MethAtlas to simulated mixtures of real WGBS data (Page 8) and to our real cfDNA from ALS and pregnancy samples (Page 10 and Page 11). We find that on simulated WGBS data, CelFiE is more accurate, with an average Pearson's correlation between the true and estimated cell-type proportions of 0.83 ± 0.16 , whereas MethAtlas had an average correlation of 0.43 ± 0.24 (Page 8). In cfDNA, we note that MethAtlas recapitulated our findings of an

elevated cfDNA component from placenta in pregnant women and higher skeletal muscle cfDNA component in ALS patients. However, it overestimates the proportion of cfDNA originating from white blood cells, likely because it does not estimate an unknown component. We believe these examples further illustrate the reasons a user would choose CelFiE when decomposing cfDNA read count data in real-world applications.

Overall, we hope to communicate to our readers that CelFiE is not a direct competitor to any existing method, but rather, a novel framework that addresses previously unaddressed aspects of cfDNA decomposition.

More importantly, there is no discussion of what methodological differences there are between the two deconvolution methods (CelFiE vs. MethAtlas) and what aspects of each method may lead to their differing performance.

We add a paragraph of discussion detailing MethAtlas' model and the differences from CelFiE (Page 3). Briefly, MethAtlas implements non-linear least squares programming and is designed to estimate cell type proportions in the context of methylation array data. That is, unlike CelFiE, it is not designed for low read count data. It also assumes the reference panel is an accurate representation of the cell types truly in the cfDNA. These different model assumptions mean that while both CelFiE and MethAtlas work well in high read count paradigms and when all the cell types in the cfDNA mixture are known, CelFiE outperforms MethAtlas when the data is lower read count, noisy, or the reference is incomplete.

Also, while there may be no other papers that are quite comparable to the CelFiE WGBS application, there are certainly many other papers on cell type deconvolution (for example, deconvolution from RNA-seq data, etc.) that from a methods perspective would make for a very interesting comparison. Have the authors explored other deconvolution methods (least squares regression, non-negative matrix factorization, support vector regression, etc.)

We agree that more comprehensive comparisons strengthen our paper. Therefore, we explore two additional methods and adapt them for cfDNA WGBS data. Firstly, we used an adaptation of least squares regression since it is a popular choice for decomposition. We find that the least-squares regression performs worse than CelFiE, especially in the context of low read depths (Figure S1A).

This method, however, is not a constrained optimization method and thus isn't directly comparable to CelFiE, which we acknowledge in our manuscript. We also developed a novel second comparison method, which we refer to as our "projection method." This method, to our knowledge, is the first application of an L1-ball projection method (Duchi et. al. 2008) to genomics or, specifically, cell-free DNA decomposition. In our approach, we optimize a binomial log-likelihood model of read count data and use an L1 projection to constrain the estimates of cell-type proportions to lie on the probability simplex. Our

projection method directly addresses some of the key constraints addressed in CelFiE: modeling read count data and having the estimates of the cell type proportions sum to one. This makes it the best available comparison to CelFiE. Our projection method performs well in low read count scenarios (Figure S1B), however, it is still unable to decompose mixtures with missing data (Figure S5), which we believe is further evidence for the utility of CelFiE.

Overall, we agree that there are several other deconvolution methods that would have made interesting comparisons, coming both from genomics and from the statistical literature more broadly. We also agree that we have the responsibility to demonstrate the improvements of our method over the state-of-the-art methods. This is why we originally chose to focus our comparisons on MethAtlas, since off-the-shelf methods generally require substantial tailoring to a specific application, such as modeling sparsity in RNA-seq data or designing methods for genotype data in population genetics. We try to do the same here with cfDNA by accommodating missing cell types and low coverage data.

and why did they settle on EM?

We added context for our decision to use an EM algorithm to optimize our model (Page 2). Overall, an EM is an extremely flexible technique to solve our likelihood, especially when we expect there to be missing data. Indeed, EM was designed as a general-purpose optimization method in the presence of missing data (in our case, the cell type of origin). In principle, other optimization tools could be used, and while they may increase speed, they would be expected to yield identical results to the EM. We also note that EM is a method, not a model, hence is not directly comparable to approaches like NMF or our above-described least-squares approach (both of which are not proper data models but do, implicitly, define error models through their objective functions).

Specific comments:

1. In simulated data, there appears to be quite a drastic difference between the performance of MethAtlas and CelFiE. It would be useful to describe the methodological improvements made in CelFiE vs. MethAtlas that could attribute to this difference. I also worry that these differences could be related to how the data was simulated instead of true methodological improvements; how can you convince the reader otherwise?

We added information on CelFiE and MethAtlas' different methodologies on Page 3. Specifically, we write,

“We also compared CelFiE to a previously published cfDNA decomposition tool, MethAtlas. Unlike CelFiE, which explicitly models WGBS reads, MethAtlas is designed to decompose methylation array data. MethAtlas also does not model missing data or estimate the methylation values for

the reference cell types. Briefly, it optimizes $\|Y\alpha - \beta\|$ using non-negative least squares constrained by $\alpha \geq 0$, where Y is a reference matrix of array data, β is the observed cfDNA methylation measured on an array, and α is the cell type proportions vector that is being solved for.”

To compare the performance between MethAtlas and CelFiE further, we ran MethAtlas on WGBS data (which is not entirely simulated, but rather mixtures of real-world data) and the ALS/pregnancy cfDNA data. We find that MethAtlas performs worse than CelFiE on WGBS, which is expected since WGBS data do not meet the specific assumptions of MethAtlas. MethAtlas does find a similar placenta signal in pregnant women (Page 10) and skeletal muscle in ALS patients (Page 11).

However, we want to emphasize that these differences in performance are related to the type of data being used. As addressed earlier, MethAtlas was not designed for low-count read data. When we simulate data closer to the ideal conditions for MethAtlas (Figure S6A), the performance is comparable to CelFiE.

2. There is a lot of focus on simulated data sets, but I think it'd be much more interesting and a better proof of concept to lead with true data applications. I found the real data examples to be much more convincing as for why one should use the CelFiE method.

We agree. This is why our twin real data examples are the heart of our paper: the pregnancy data are a positive control, and the ALS data show how our method can translate into real clinical utility. In our new writing, we hope that our simulations convey the key parameters of real data that govern the utility of CelFiE, which would make it more trustworthy to clinical stakeholders. Therefore, we ask the reviewer to consider allowing us to leave the paper structure as is. However, we added another sentence to the intro about the real data results so that the readers with a similar opinion will know that real data applications are presented further down the paper (Page 2).

3. When varying the number of CpG sites in the simulated data sets, were these CpG sites randomly chosen or were only the top 100, 1000, etc. most informative/variable sites chosen? I imagine specifically picking more informative sites will increase performance especially when using only a small number (100) of sites. In general, selecting appropriate CpG sites can have a large impact on deconvolution accuracy. It would be valuable to perform some type of sensitivity analysis showing how for example choosing different numbers of highly informative CpG sites affects the deconvolution results.

We agree with the reviewer's hypothesis that picking more informative sites increases decomposition performance. We added a supplemental figure to demonstrate this (Figure S7). We simulated data for 100, 1000, and 10000 CpGs across 10 cell types. The methylation value for the 10 cell types at a given CpG was drawn from a normal distribution centered on 0.5, with variances ranging from 0.01 to 1. Thus, CpGs with low

variance would give very little information about cell type, since all 10 cell types at that CpG would have nearly the same methylation value. As expected, we find that as we increase the informativeness and increase the number of sites, the performance of CelFiE improves.

Reviewer #2 (Remarks to the Author):

The manuscript by Caggiano et al. entitled "CelFiE: Comprehensive cell type decomposition of circulating cell-free DNA" develop a method to use WGBS data for low coverage data in cell free DNA in blood named CelFiE (CELI Free DNA Estimation via expectation-maximization). Then they applied this method to investigate cell-free data in pregnant vs non-pregnant women and patients with amyotrophic lateral sclerosis (ALS) vs. age-matched controls. The goal of this work is to overcome several limitations posed by the limited information in microarrays and the small amounts of DNA that are required to be captured in cell-free DNA analyses. An important feature that they underline is how the model is resistant to heavy-tailed distributions and potential model-assumptions. The assumptions in the model and the flexibility of adding unknowns increase the potential applications for this methodology.

We thank the reviewer for their encouraging comments on CelFiE and their positive assessment of its potential applicability to the community.

Said that, I would like some clarifications about your results:

Comment 1: pages 2-3 section 2.2: Although I understand your idea to replicate and compare to Meth_atlas, the sentence is not precise. There are no 25 cell types, instead Moss et al. use both isolated cell-types and tissues, most of them from cancer patients. You may want to adapt your sentence to be more precise. This is mentioned later on section 2.3

We have improved our language on Page 3 to clarify that Moss et. al. have a reference panel composed of both cell types and tissues.

Comment 2: page 4 paragraph 3, fig 3: You mention how the accuracy of your results are based on the coverage. What is the minimum coverage for accomplishing a 0.1%? Is it the 1000x, is it lower? Could you, theoretically, capture proportions lower than 0.1% with deeper coverage? When looking at your figure 3 there are differences versus the background and there are significant differences between the true fraction and the detected fraction. However, the median detected values are over the red line with a wide variability. Could you provide some information for the precision of your estimates?

First, we found that 1000x is the lowest depth at which we could detect a 0.1% difference. As we increased the depth to 10,000X, we could significantly detect a 0.01% difference ($p=8.32 \times 10^{-9}$). However, we note, now on Page 5, that this detection would likely be

impeded by biological constraints, such as the amount of cfDNA actually available in blood and degradation induced by bisulfite conversion.

Second, we agree that uncertainty in our estimates is non-trivial, and we have clarified and quantified this uncertainty by adding standard error bars on our estimates on Page 5.

Comment 3: on page 2 section 2.1 you mention the problem of missing cells in your reference. Then you explore one and 2 missing values (Fig S1 and Fig S2) What could be a potential problem for those missing values, if you miss one of the main cells (instead of one at random) could you quantify the bias of your results? Have you quantified the amount of expected bias in both extremes with low proportions and high proportions?

To address how missing cell types cause bias in the decomposition estimates, we add a supplemental figure that contrasts CelFiE's estimates with and without an unknown component (Figure S4). We do this for a mixture where there is an abundant missing cell type (meaning it is the "main" or dominant cell type in the mixture) and a mixture where the missing cell type is close to the average. We then show that when there is no unknown estimation, decomposition estimates for other tissues will be increased by 25 percent on average across all cell types. When CelFiE is run with an unknown component, estimates are more accurate, with only a 2-10% difference from the true cell type proportion on average. Furthermore, when we compare CelFiE's predictions to those of MethAtlas, linear least squares regression, and our projection method outlined on Page 7, we find that these methods also overestimate the true cell type proportion (Figure S5), which we believe motivates the use of CelFiE and its unknown estimation.

You show in the figures some outliers that had correlation below 0.5 and even close to zero.

This is likely a consequence of the different data being drawn in each simulation replicate. We now state this in the text on page 7.

Further, it is not shown the specific tissue/cell-type in your comparison between your method and MethAtlas. Were these results different in specific tissues or were similar independent of the tissue? Please expand on those results.

We now clarify in the text that the original MethAtlas/CelFiE comparisons were done in completely simulated data. However, to more adequately compare MethAtlas and CelFiE, we ran MethAtlas on our simulated mixtures of WGBS (Page 8). We found that MethAtlas performed worse WGBS data. Specifically, the average Pearson's correlation between true and estimated cell-type proportions for MethAtlas was 0.43 ± 0.23 . For CelFiE, ran on the same data, the average Pearson's correlation was 0.86 ± 0.16 .

Additionally, we ran MethAtlas on our real cfDNA datasets (Page 10 and Page 11). In general, MethAtlas recapitulated our main findings in these datasets. Namely, it estimated a higher proportion of cfDNA originating from the placenta in pregnant women and an increased proportion of cfDNA originating from skeletal muscle in ALS patients. Notably, CelFiE estimates a more significant difference in the skeletal muscle proportion between cases and controls, $p = 2.36 \times 10^{-3}$, versus $p=1.25 \times 10^{-2}$. We believe this adds further context to the difference in our two methods.

Comment 4: page 7: in addition to the previous comment you mention a high correlation of the true proportions with you method around the mean of the estimation. I do not see any estimates of the precision of your estimates? How precise were those estimations? How the estimate precision was affected with an increased amount of unknown cells in Fig. 5.

We add S.E. on all Pearson's correlations and means reported.

Comment 5: Fig 7: Could you add some information about the unknown fraction observed (>20%) in both the pregnant and non-pregnant women?

To explore this, we perform hierarchical clustering of the estimated methylation values of the unknown component in our pregnancy and ALS CelFiE decompositions with the true reference methylation proportions (Figures S11 and S16). In the pregnancy data, we find that the unknown clustered most closely with endothelial cells, suggesting that improved reference panels could add additional biological information. In the ALS data, we find that the unknown does not cluster closely with any specific tissue, suggesting that it may be a combination of several missing tissues. Together, these examples demonstrate two scenarios where the unknown component estimated by CelFiE can provide useful information to a cfDNA researcher.

Comment 6: Fig S8: could you expand about the other components in your deconvolution and their biological meaning? Dendritic cells, small intestines, etc. Are those target cells/organs for the disease? Are those random findings?

We added context on Page 10 and 11 detailing why we expect a high proportion of white blood cell types and that other detected tissues, such as the small intestine or stomach, are likely due to the high turnover rate of cells in those tissues.

Comment 7: How is your model handling very similar cell/tissue types? You mention something about small intestine and other GI tissues, are these accurate predictions? are general approximations? Could you add some information about how to contextualize those findings?

This is an excellent point that we wish to address in future work by explicitly modeling the relationships between tissues and cell types in our reference. However, to

demonstrate this in our current framework, we simulated a mixture of 10 cell types. Two of these cell types had identical methylation proportions. Next, we added normally distributed noise to one of the cell types. The variance of the noise was set to be between 0.0 and 1.0. We found that as the noise increased, meaning that the two cell types were becoming less similar, CelFiE's ability to identify the correct cell type proportion increased. This result is in Figure S2.

Reviewers' Comments:

Reviewer #1:

Remarks to the Author:

I thank the authors for providing extensive responses to my questions and comments. They have been largely addressed.

Thank you for adding extra model comparisons (least squares regression and L1-ball) and adding some context on your choice of probabilistic modeling. I am still a bit concerned on the over-reliance of simulated data when it comes to reporting CelFiE's performance compared to these other models—could you add comparisons to these extra models in the real data sets as well? It will be important to show that performance comparisons from simulated data carry over to real data as well.

Reviewer #2:

Remarks to the Author:

The manuscript by Caggiano et al. entitled "CelFiE: Comprehensive cell type decomposition of circulating cell-free DNA" develop a method to use WGBS data for low coverage data in cell free DNA in blood named CelFiE (CELL Free DNA Estimation via expectation-maximization). Then they applied this method to investigate cell-free data in pregnant vs non-pregnant women and patients with amyotrophic lateral sclerosis (ALS) vs. age-matched controls. The goal of this work is to overcome several limitations posed by the limited information in microarrays and the small amounts of DNA that are required to be captured in cell-free DNA analyses. An important feature that they underline is how the model is resistant to heavy-tailed distributions and potential model-assumptions. The assumptions in the model and the flexibility of adding unknowns increase the potential applications for this methodology.

After the first round of comments the authors have addressed all my concerns and added important information for the readers. I only have one additional minor comment: Could you please add the dbgap identifier or hyperlink for your data?

We thank the reviewers for their further comments and were glad to see that they believe we have added important information for readers after the first round of reviews. We address each of the additional comments in the following document. We believe our manuscript is improved as a result.

Reviewer comments are in plain text below with our responses in bold.

Reviewer #1 (Remarks to the Author):

I thank the authors for providing extensive responses to my questions and comments. They have been largely addressed.

Thank you for adding extra model comparisons (least squares regression and L1-ball) and adding some context on your choice of probabilistic modeling. I am still a bit concerned on the over-reliance of simulated data when it comes to reporting CelFiE's performance compared to these other models— could you add comparisons to these extra models in the real data sets as well? It will be important to show that performance comparisons from simulated data carry over to real data as well.

We ran both our implementation of least squares regression and our L1 projection method on the real cfDNA taken from pregnant women (Figure S12) and our ALS case-control study (Figure S16). The results are similar to those produced by MethAtlas, with large estimates for cell types known to be rare in blood. This is consistent with our simulated data when unknown cell types were not adequately modeled (Figure S12D and S16D).

Before going into detail, we highlight a couple points. First, the least-squares regression implementation and L1-projection implementation both leverage the backbone of QC, site choice, and preprocessing contained in CelFiE. Off-the-shelf versions of linear regression will perform substantially worse. Additionally, the projection method is the first, to our knowledge, application of the Duchi et al. L1-ball projection to this biological problem. Therefore, both of these adaptations to the problem of cfDNA deconvolution should be considered our novel methods even though they use pre-existing models. See for example (Chakravarthy et al., 2019, Nature Communications; Rahmani et al., 2015, Nature Methods, Chiu et al., 2019, BMC Genomics; Qiao et al., 2012, PLoS Computational Biology), all of which use existing models, but do considerable work to adapt them to new problems. Second, we agree with the reviewer that real data is the ultimate test case. In this situation, however, there is no ground truth available. Therefore, we rely on statistical features of the regression and known biology to examine performance. These additions are now on pages 11 and 12.

As expected, linear least squares performed poorly, producing negative estimates, illustrating a major drawback of using this approach. Furthermore, least squares, the projection method, and MethAtlas estimated proportions of blood cell types that are

inconsistent with known cell-type proportions in whole blood (Figure S12 A-C). For example, all three methods estimated a high proportion of erythroblasts, around 24%. This is unlikely to be true given that erythroblasts are a relatively rare cell type in adult blood (Constantino et al 2000). While we acknowledge that this proportion may be tagging another red blood cell precursor not well captured in our current reference, it also may be a bias introduced by missing tissues in the reference panel. To illustrate this, we ran CelFiE with and without an unknown component on the same data (Figure S12D). With an unknown, CelFiE estimated an average erythroblast proportion of 7.3%. This increased to 29% when CelFiE was ran without an unknown, suggesting that the unknown component of our method may be important in real data.

In the context of ALS, we similarly find that least squares regression, the projection method, and MethAtlas have high estimates of erythroblasts that are consistent with estimates produced by CelFiE ran without an unknown (Figure S16). Additionally, linear least squares regression again produced negative estimates. We believe this is further evidence that CelFiE may be a desirable method for the decomposition of real cfDNA data.

Reviewer #2 (Remarks to the Author):

The manuscript by Caggiano et al. entitled "CelFiE: Comprehensive cell type decomposition of circulating cell-free DNA" develop a method to use WGBS data for low coverage data in cell free DNA in blood named CelFiE (CELL Free DNA Estimation via expectation-maximization). Then they applied this method to investigate cell-free data in pregnant vs non-pregnant women and patients with amyotrophic lateral sclerosis (ALS) vs. age-matched controls. The goal of this work is to overcome several limitations posed by the limited information in microarrays and the small amounts of DNA that are required to be captured in cell-free DNA analyses. An important feature that they underline is how the model is resistant to heavy-tailed distributions and potential model-assumptions. The assumptions in the model and the flexibility of adding unknowns increase the potential applications for this methodology.

After the first round of comments the authors have addressed all my concerns and added important information for the readers. I only have one additional minor comment: Could you please add the dbgap identifier or hyperlink for your data?

We uploaded the raw and processed data for our whole-genome bisulfite sequencing ALS study to NCBI GEO, so that it can be freely accessible. We add the identifier GSE164600 to the data availability section of our paper (Page 17).

Reviewers' Comments:

Reviewer #2:

Remarks to the Author:

The authors have addressed all the comments from the reviewers. I do not have any additional comments.